. Pathogens

# BRD4 modulates antimicrobial defense via non-canonical NRF2 activation in macrophages to confer protection against sepsis

Jinfeng Hu[1]*☯, Xuming Gao[1]☯, Guo Li[1]☯, Xiaoxin He[1]☯, Yanting Ke[2]☯, Zhongheng Zhang[3,4,5]☯, Duozhi Pang[1], Zhen Lin[1], Cailian Xie[1], Xiaoting Chen[6], Minting Jiang[1], Shuping Zheng[7], Suhong Yu[6], Mingrui Lin[8], Dun Pan[9], Xiaopei Shen[2]*, Xiangming Hu [ID][1]*

**1** Fujian Key Laboratory of Cognitive Function and Diseases, Institute for Basic Medical Sciences, School of Basic Medical Sciences, Fujian Medical University, Fuzhou, China, **2** Department of Bioinformatics, Fujian Key Laboratory of Medical Bioinformatics, Institute of Precision Medicine, School of Medical Technology and Engineering, Fujian Medical University, Fuzhou, China, **3** Department of Emergency Medicine, Provincial Key Laboratory of Precise Diagnosis and Treatment of Abdominal Infection, Sir Run Run Shaw Hospital, Zhejiang University School of Medicine, Hangzhou, China, **4** School of Medicine, Shaoxing University, Shaoxing, China, **5** Longquan Industrial Innovation Research Institute, Lishui, China **6** Fujian Provincial Key Laboratory of Cancer Metastasis Chemoprevention and Chemotherapy, Fuzhou University, Fuzhou China, **7** Public Technology Service Center, Fujian Medical University, Fuzhou, China, **8** Intensive Care Unit, The Affiliated People's Hospital of Fujian University of Traditional Chinese Medicine, Fuzhou, China, **9** Department of Gastrointestinal Surgery, The First Affiliated Hospital of Fujian Medical University, Fuzhou, China

☯ These authors contribute equally to this work.
* xmhu2003@fjmu.edu.cn (XMH), hujinfeng@fjmu.edu.cn (JFH) xshen@fjmu.edu.cn (XPS)

## Abstract

Sepsis is a life-threatening condition characterized by dysregulated immune responses and high mortality, driven by persistent pathogens and compromised antimicrobial defenses. We identify BRD4, an epigenetic regulator, as a crucial modulator of macrophage antimicrobial function and survival in sepsis. Sepsis significantly reduces BRD4 expression in monocytes/macrophages in both human patients and murine models, with decreased *BRD4* levels correlating with disease severity. Myeloid-specific deletion of *Brd4* exacerbates mortality by impairing macrophage phagocytosis and bactericidal activity. BRD4 interacts with NRF2, disrupting the NRF2-KEAP1 complex, which enhances NRF2 stability and nuclear translocation, leading to the upregulation of scavenger receptors essential for bacterial clearance. Notably, restoration or activation of NRF2 rescues the macrophage functional defects induced by Brd4 deficiency both *in vitro* and *in vivo*, highlighting the therapeutic potential of this pathway. Our findings reveal that BRD4 downregulation in human sepsis predicts disease severity, presenting BRD4 as both a biomarker and a therapeutic target. The BRD4-NRF2 axis offers a novel approach to restoring host defense and improving sepsis treatment strategies.

**Data availability statement:** All relevant data are within the manuscript and its Supporting Information files.

**Funding:** This work was supported by the National Natural Science Foundation of China (81902842 to X.M.H. and 81801974 to J.F.H.), the Joint Funds for Innovation in Science and Technology, Fujian Province (2023Y9001 to X.M.H.), the Natural Science Foundation of Fujian Province (2025J01690 to X.M.H. and 2025J01692 to J.F.H.), and Startup Fund for scientific research, Fujian Medical University (2023QH2004 to X.M.G). The funders had no role in study design, data collection and analysis, decision to publish, or preparation of the manuscript.

## Author summary

Sepsis lethality is driven by impaired macrophage function, but the underlying mechanisms are unclear. We found that BRD4 expression is significantly reduced in monocytes/macrophages from septic patients and mice, and this reduction correlates with disease severity. Mechanistically, BRD4 interacts with NRF2 to disrupt its binding with KEAP1, thereby enhancing NRF2 stability and its transactivation of scavenger receptors crucial for bacterial clearance. Consequently, myeloid-specific Brd4 deletion exacerbates sepsis by compromising macrophage phagocytosis and bactericidal activity. Importantly, restoration or activation of NRF2 rescues these defects and improves survival. Our study identifies BRD4 downregulation as a key event in sepsis pathogenesis, highlighting the BRD4-NRF2 axis as a promising therapeutic target and BRD4 as a potential biomarker for disease severity.

## Introduction

Sepsis is a leading cause of death in critically ill patients, affecting 48.9 million people annually and causing up to 11 million deaths [1,2]. Pathogen persistence and excessive inflammation drive multiorgan dysfunction and high mortality [3]. While antibiotics are frequently used, their effectiveness is limited by resistance and poor drug delivery [4]. Additionally, cytokine-targeted therapies have largely failed in clinical trials, as they cannot fully mitigate sepsis-related damage in the presence of persistent pathogens [5,6]. Recent studies highlight that sepsis-induced immune dysfunction impairs pathogen clearance [7–9], emphasizing the need for a deeper understanding of immune cell defects to guide novel therapeutic approaches.

Macrophages are critical in all phases of sepsis, with their distribution spanning peripheral blood and organs such as the lungs, liver and kidneys [10,11]. Early in sepsis, they activate as the first line of defense against pathogens. However, as sepsis progresses to an immunosuppressive phase, macrophage functions including phagocytosis, bactericidal activity, antigen presentation, and cytokine secretion, are severely impaired, leading to impaired antibacterial responses and increased mortality [12–14]. Emerging evidence links macrophage dysfunction to epigenetic changes that modulate immune responses in sepsis [15,16], highlighting epigenetic regulators as potential therapeutic targets.

BRD4 (bromodomain-containing protein 4) is a crucial epigenetic and transcriptional regulator in response to stressors, including inflammation and microbial infections [17–19]. By binding acetylated histones and non-histone proteins through its bromodomains, and recruiting transcription activators via its extra-terminal and C-terminal domains, BRD4 drives inflammatory cytokine and chemokine expression in macrophages [20]. Inhibition of BRD4 reduces inflammatory gene expression and protects mice from endotoxin-induced sepsis [21]. However, myeloid-specific *Brd4* knockout mice, while resistant to endotoxemia, exhibit heightened susceptibility to

bacterial infections, suggesting impaired innate immunity [21,22]. BRD4 expression varies across immune cells during different infections, being upregulated in macrophages infected with *Leishmania donovani* and downregulated in memory CD4+ T cells during HIV-1 infection [23,24]. While these findings highlight BRD4's role in innate immunity, its exact function and regulatory mechanisms in sepsis remain poorly understood.

Nuclear factor erythroid 2-related factor 2 (NRF2) is a key regulator of redox homeostasis and innate immune responses against microbial infections [25,26]. It modulates inflammatory responses triggered by Toll-like receptors (TLRs) upon pathogen recognition, preventing excessive inflammation and subsequent tissue damage. When inflammation intensity becomes dysregulated, TLR signaling triggers the autophagic degradation of NRF2's negative regulator KEAP1 (Kelch-like ECH-associated protein 1), activating NRF2 and inhibiting NF-κB and NLRP3 inflammasome-mediated inflammation [27,28]. NRF2 also enhances the expression of macrophage-specific phagocytic receptors, such as MARCO (macrophage receptor with collagenous structure) and MSR1 (macrophage scavenger receptor 1), boosting their antibacterial functions [29]. In a cecal ligation and puncture (CLP)-induced sepsis model, myeloid-specific *Nrf2* knockout mice show increased mortality due to immune dysregulation and impaired bacterial clearance [30]. Despite its recognized role, the mechanisms governing NRF2 activation during sepsis remain poorly understood.

Here, we observed decreased BRD4 expression in monocytes/macrophages from septic patients and mice. Myeloid-specific BRD4 deficiency disrupts phagocytosis and bactericidal activity of macrophages, significantly reducing septic mouse survival. BRD4 enhances NRF2 protein stability and nuclear translocation via a non-canonical mechanism, in which BRD4 binds to NRF2, disrupting the interaction between NRF2 and KEAP1. This process ultimately promotes the expression of macrophage scavenger receptors, such as MARCO and MSR1. Targeting NRF2 with sulforaphane enhances MARCO and MSR1 expression, improving bacterial clearance in sepsis. Our findings reveal a defective BRD4-NRF2-MARCO/MSR1 pathway that impairs macrophage antimicrobial function in sepsis, offering new insights into immune dysfunction and potential therapeutic strategies for sepsis.

## Results

### BRD4 is downregulated in monocytes/macrophages during Sepsis

BRD4 plays a dual role in innate immunity: myeloid-specific Brd4 deletion increases mortality during bacterial infections, while conferring protection against lipopolysaccharide (LPS)-induced cytokine storm in mice [21,22]. Given that bacterial sepsis involves both bacterial invasion and excessive inflammation, we sought to investigate the role of BRD4 in its pathophysiology. To address this, we first evaluated BRD4 expression across human and murine sepsis models. Bulk RNA-seq and microarray data from septic patients revealed consistent downregulation of *BRD4* transcripts (Fig 1A), significant after adjusting for age and sex (S1 Table). Receiver operating characteristic analysis demonstrated BRD4's robust ability to distinguish septic patients from healthy controls (Fig 1B). Longitudinal analysis showed that *BRD4* expression was lowest at sepsis onset (day 1) but significantly increased by day 3 and continued rising through day 5 (S1A Fig).

Given the cellular heterogeneity of peripheral blood and the potential cell type-specific functions of BRD4, we sought to identify which immune cell populations exhibited *BRD4* expression patterns correlating with sepsis progression. Leveraging the well-established clinical utility of C-reactive protein (CRP) and procalcitonin (PCT) as sepsis biomarkers [31], we performed correlation analyses between *BRD4* expression and these inflammatory markers using scRNA-seq data from septic patient blood. Our analysis uncovered a striking, monocyte-specific pattern: *BRD4* mRNA levels in monocytes showed a strong negative correlation with both serum CRP and PCT levels (Fig 1C, 1D). This inverse relationship was absent in neutrophils and other peripheral blood mononuclear cells (PBMCs), including T cell and B cell populations (S1B and S1C Fig). Importantly, comparative scRNA-seq analysis demonstrated consistent downregulation of *BRD4* in septic patient monocytes relative to healthy controls, with this suppression maintained across varying degrees of disease severity (Fig 1E).

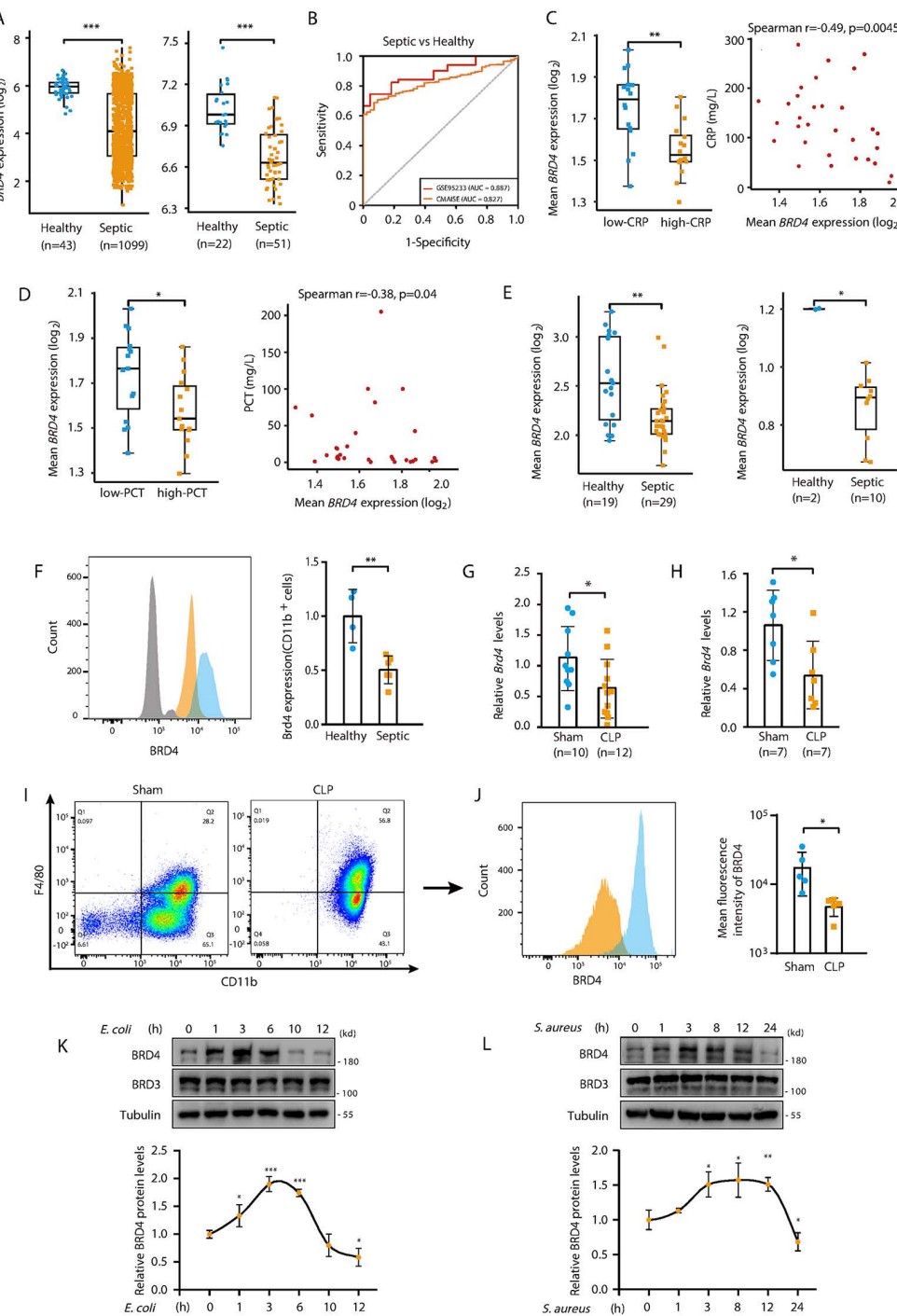

**Fig 1. BRD4 is downregulated in monocytes/macrophages during Sepsis. (A)** Comparison of *BRD4* mRNA levels (log2 transformed) in whole blood from healthy controls and septic patients. Data sources: CMAISE (Chinese Multi-omics Advances in Sepsis) (left); GSE95233 (right)). **(B)** Receiver operating characteristic (ROC) analysis of *BRD4* expression for the diagnosis of sepsis. **(C)** Comparison of *BRD4* mRNA levels in monocytes between low-CRP and high-CRP groups. The cutoff for binary classification was set at the median CRP levels in septic patients (left). Scatter plot showing the correlation between *BRD4* mRNA levels in monocytes (x-axis) and CRP levels (y-axis) (right). Each point represents an individual patient sample. Data sources: CMAISE. **(D)** Analysis of PCT levels (other parameters as described in **B**). Data sources: CMAISE. **(E)** Comparison of *BRD4* mRNA levels (log-normalized counts) in monocytes from healthy controls and septic patients. Data sources: SCP548 (left); GSE167363 (right). **(F)** Flow cytometric

analysis of BRD4 protein levels in monocytes from the peripheral blood of healthy controls (n = 4) and septic patients (n = 6). **(G&H)** *Brd4* mRNA levels in WT mice peritoneal cells (G) or peritoneal macrophages (H) 24 hours after sham or CLP surgery, as determined by qPCR (n = 7-12). **(I)** Flow cytometry analysis of peritoneal macrophages from WT mice 24 hours after sham or CLP surgery. **(J)** Flow cytometric detection of BRD4 protein levels in peritoneal macrophages from WT mice 24 hours after sham or CLP surgery (n = 5). **(K & L)** Western blot assay of BRD4 protein levels in BMDMs at different times after induction by *E. coli* (MOI = 5) **(K)** or *S. aureus* (MOI = 10) **(L)**. Quantification results are shown (n = 3). Data presented in panels A–E are expressed as medians ± interquartile range and were analyzed using the Wilcoxon rank-sum test (two-tailed). Other data are shown as means ± standard deviation (SD). Statistical analyses were performed using two-way ANOVA with Tukey's post-hoc test for multiple comparisons. *$p < 0.05$, **$p < 0.01$, ***$p < 0.001$.

To further validate these findings, we performed flow cytometry analysis, which confirmed a marked reduction in BRD4 protein levels in monocytes isolated from the peripheral blood of septic patients (Fig 1F). Similarly, peritoneal cells, including peritoneal monocytes/macrophages, from mice with CLP-induced polymicrobial sepsis also exhibited a significant decrease in *Brd4* mRNA expression compared to sham-operated controls (Fig 1G, 1H). Since macrophages play a central role in the innate immune response, we next assessed BRD4 protein levels in peritoneal macrophages from septic mice. Consistent with the above findings, BRD4 protein levels were significantly reduced in peritoneal macrophages from septic mice compared to controls (Fig 1I, 1J).

Finally, we evaluated BRD4 expression in murine *in vitro* sepsis models. Bone marrow-derived macrophages (BMDMs) were stimulated with either *Escherichia coli* (gram-negative) or *Staphylococcus aureus* (gram-positive). In both models, we observed that BRD4 mRNA and protein levels initially increased during the early stages of bacterial infection but subsequently declined, whereas BRD3 expression remained unchanged (Figs 1K, 1L and S1D). Together, these results suggest that BRD4 is critically involved in the immune regulation of macrophages during sepsis, with its expression being dynamically modulated in response to bacterial infection.

## Myeloid lineage–specific *Brd4* knockout (*Brd4*-CKO) mice are more susceptive to CLP-induced polymicrobial sepsis

To investigate the role of macrophage BRD4 in sepsis, myeloid-specific Brd4-deficient (*Brd4*-CKO) mice were generated by crossing *Brd4* floxed mice with Lyz2-Cre mice [32,33]. When *Brd4*-CKO mice were subjected to CLP, we observed a significant increase in mortality compared to septic wild-type (WT) littermates (Fig 2A).

Lethality in sepsis is linked to multiple organ failure. Therefore, we then investigated the histopathological changes or function of major organs (such as liver, lung and spleen) often injured in patients with sepsis. CLP-treated WT mice displayed liver injury with inflammatory cell infiltration and clotting, which was exacerbated in *Brd4*-CKO mice (Fig 2B). Serum levels of AST and ALT, markers of liver injury, were also higher in *Brd4*-CKO mice (Fig 2C). Lung damage was more severe in *Brd4*-CKO mice, with increased hemorrhage, alveolar thickening, and epithelial damage (Fig 2D). Consistent with these findings, lung injury scores and myeloperoxidase (MPO) levels were elevated in *Brd4*-CKO mice (Fig 2E, 2F). The spleens of septic *Brd4*-CKO mice showed increased lipid vacuolization and apoptosis compared to WT mice (Figs 2G and S2A). Collectively, these results suggest that *Brd4*-CKO mice experience higher mortality due to more severe multiple organ failure.

To gain further insight into the clinical relevance of these observations, we examined BRD4 expression in monocytes from septic patients. The Sequential Organ Failure Assessment (SOFA) score is widely used for diagnosing sepsis and assessing the extent of organ dysfunction [34]. Stratifying septic patients by their median SOFA score revealed a significant inverse correlation between SOFA scores and *BRD4* expression in monocytes (Fig 2H). Specifically, monocytes from patients with higher SOFA scores exhibited markedly reduced *BRD4* expression compared to those from patients with lower SOFA scores (Fig 2I). This relationship was unique to monocytes, as no significant correlation was observed in whole blood or other PBMCs (S2B and S2C Fig). These findings further support the notion that BRD4 expression in monocytes is associated with the severity of organ dysfunction in septic patients.

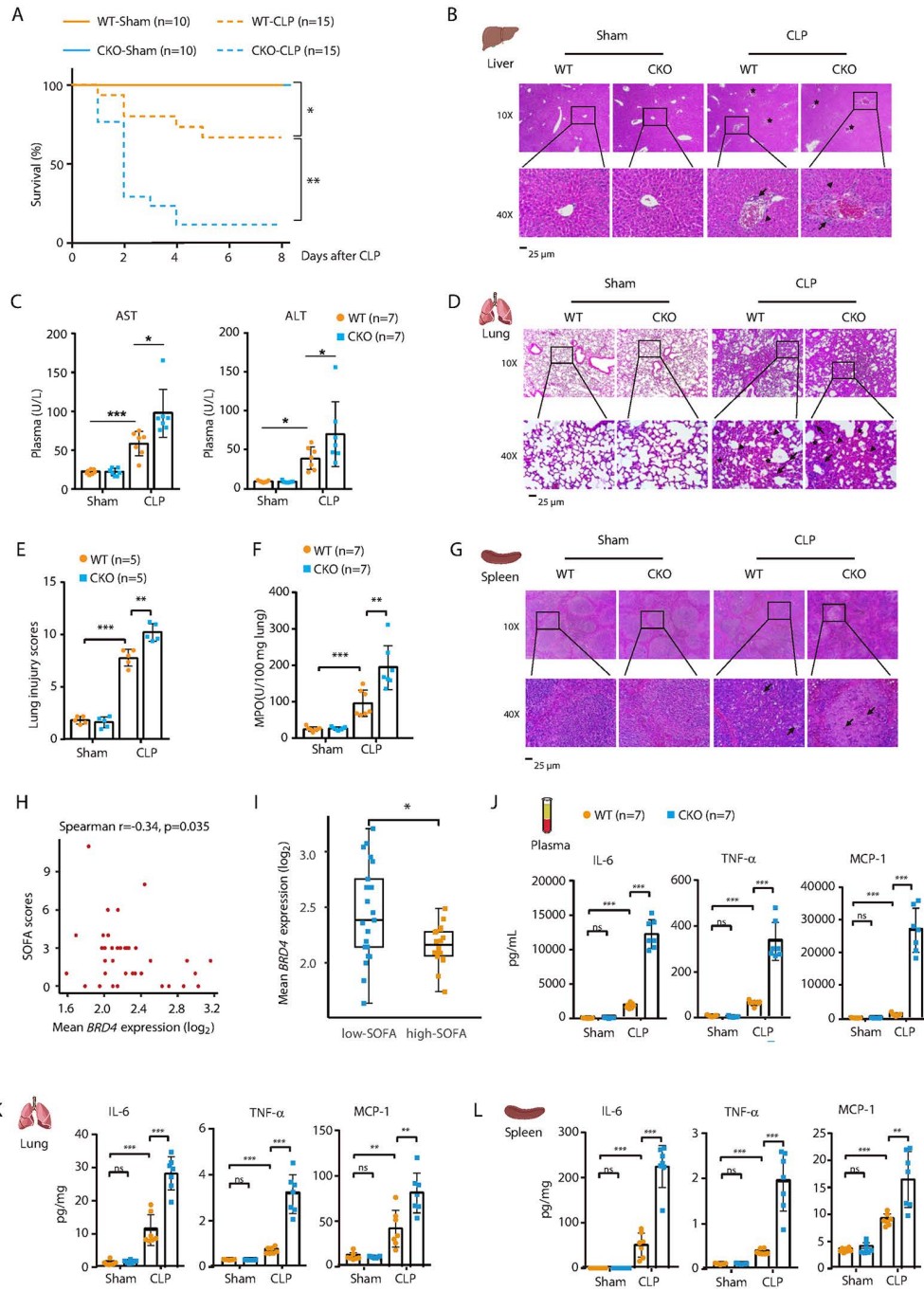

**Fig 2. *Brd4*-CKO mice exhibit increased susceptibility to CLP-induced polymicrobial sepsis. (A)** Survival curves of WT and *Brd4*-CKO mice following sham or CLP surgery over 8 days (n = 10-15). **(B)** Representative hematoxylin and eosin **(H&E)**-stained liver tissue sections from WT and Brd4-CKO mice (n = 5), 24 hours after sham or CLP surgery. Black arrows indicate infiltration of inflammatory cells, arrowheads denote congested blood sinusoids, and stars indicate areas of hyperemia. **(C)** Plasma levels of aspartate aminotransferase (AST) and alanine aminotransferase (ALT) in mice treated as described in panel **B**. **(D)** Representative H&E-stained lung tissue sections from mice treated as described in panel **B**. Black arrows indicate inflammatory cell infiltration, arrowheads show reduced alveolar air space, and stars indicate intra-alveolar capillary hemorrhages. **(E)** Lung injury severity in mice treated as described in panel **B**. Lung injury was assessed using a lung injury scoring system (n = 5). **(F)** Pulmonary myeloperoxidase (MPO) activity in the lungs of mice treated as described in panel B (n = 7). **(G)** Representative H&E-stained spleen tissue sections from mice treated as described in panel B (n = 5). Black arrows indicate apoptotic splenocytes with pyknotic and karyorrhectic nuclei. (H & **I)** Scatter plot showing the correlation between

*BRD4* mRNA levels in monocytes (x-axis) and Sequential Organ Failure Assessment (SOFA) scores (y-axis) **(H)**. Comparison of BRD4 mRNA levels in monocyte between Low-SOFA and High-SOFA groups **(I)**. The cutoff for binary classification was set at the median SOFA scores in patients. Data source: SCP548. **(J-L)** Quantification of IL-6, TNF-α, and MCP-1 levels in plasma **(J)**, lung **(K)**, and spleen (L) from mice treated as described in panel B (n = 7). Data presented in panels H & I are expressed as medians ± interquartile range and were analyzed using the Wilcoxon rank-sum test (two-tailed). Other data are shown as means ± SD. Statistical analyses were performed using two-way ANOVA with Tukey's post-hoc test for multiple comparisons. *$p < 0.05$, **$p < 0.01$, ***$p < 0.001$. ns, not significant.

Sever sepsis is frequently accompanied by lethal systemic inflammation, which contributes to tissue damage and organ failure [35]. We examined inflammatory responses by measuring proinflammatory cytokines in plasma. CLP-treated *Brd4*-CKO mice exhibited higher levels of IL-6, TNF-α, and MCP-1 than WT mice (Fig 2J). In the lungs and spleens, *Brd4*-CKO mice had elevated cytokine levels, further indicating enhanced systemic and local inflammation (Fig 2K, 2L). Additionally, *Brd4*-CKO mice had significantly higher bacterial loads in the peritoneal cavity, blood, spleen, liver, and lungs (Fig 3A), suggesting impaired bacterial clearance in these mice. These findings indicate that Brd4 in myeloid cells plays a crucial role in improving survival during polymicrobial sepsis by facilitating bacterial clearance and modulating inflammation.

## BRD4 deficiency attenuates phagocytic and bactericidal activities of macrophages

Sepsis-induced mortality is largely driven by an initial surge in inflammatory cytokine production, followed by impaired microbial clearance in later stages [36]. The increased bacterial load observed in septic *Brd4*-CKO mice indicated that myeloid cells in these animals might exhibit impaired bacterial clearance. To explore whether BRD4 influences bacterial phagocytosis by macrophages, we first assessed the ex vivo phagocytic activity of WT and *Brd4*-deficient BMDMs. After incubation with GFP-labeled *Escherichia coli* (*E. coli*, Gram-negative) or *Staphylococcus aureus* (*S. aureus*, Gram-positive), *Brd4*-deficient BMDMs showed a significant reduction in their ability to phagocytose both *E. coli* and *S. aureus* compared to WT BMDMs (Fig 3B, 3C).

The role of BRD4 in macrophage phagocytosis was further examined using zymosan particles, which are composed primarily of beta-glucan and are commonly used as a model for fungal recognition and macrophage phagocytosis [37]. *Brd4*-deficient BMDMs demonstrated a marked reduction in zymosan particle ingestion compared to WT BMDMs (Fig 3D). Scanning electron microscopy (SEM) analysis revealed an abundance of zymosan particles on the surface of *Brd4*-deficient BMDMs, while transmission electron microscopy (TEM) imaging showed a reduced accumulation of intracellular zymosan particles (Fig 3E). These observations provide further evidence of impaired phagocytic activity in *Brd4*-deficient BMDMs.

Next, we evaluated the role of BRD4 in bacterial phagocytic capacity *in vivo*. GFP-labeled *E. coli* or *S. aureus* were injected into the peritoneal cavities of WT or *Brd4*-CKO mice. After 15 minutes, mice were sacrificed, and peritoneal macrophages were analyzed for phagocytic activity by flow cytometry. *Brd4*-CKO mice exhibited significantly reduced macrophage phagocytosis compared to WT mice (Figs 3F and, 3G).

Both phagocytic and bactericidal activities are crucial for host defense against bacterial infections [38]. To further investigate the bactericidal capacity of *Brd4*-deficient macrophages, we performed antibiotic protection assays (APA) using BMDMs infected with *E. coli* and *S. aureus*. *Brd4*-deficient BMDMs demonstrated significantly diminished bactericidal activity compared to WT BMDMs (Fig 3H, 3I). This reduction in bactericidal function was likely associated with decreased nitric oxide (NO) and reactive oxygen species (ROS) production (S3 Fig). Taken together, these findings indicate that the impaired bacterial clearance observed in *Brd4*-deficient macrophages is likely due to compromised phagocytic and bactericidal functions.

## Macrophages from *Brd4*-CKO mice exhibit reduced expression of MARCO and MSR1

To explored the mechanism underlying defect antibacterial ability in *Brd4*-deficient macrophages, we performed global RNA-seq analysis of WT and *Brd4*-deficient BMDMs. The analysis was performed after exposure (or no exposure) to

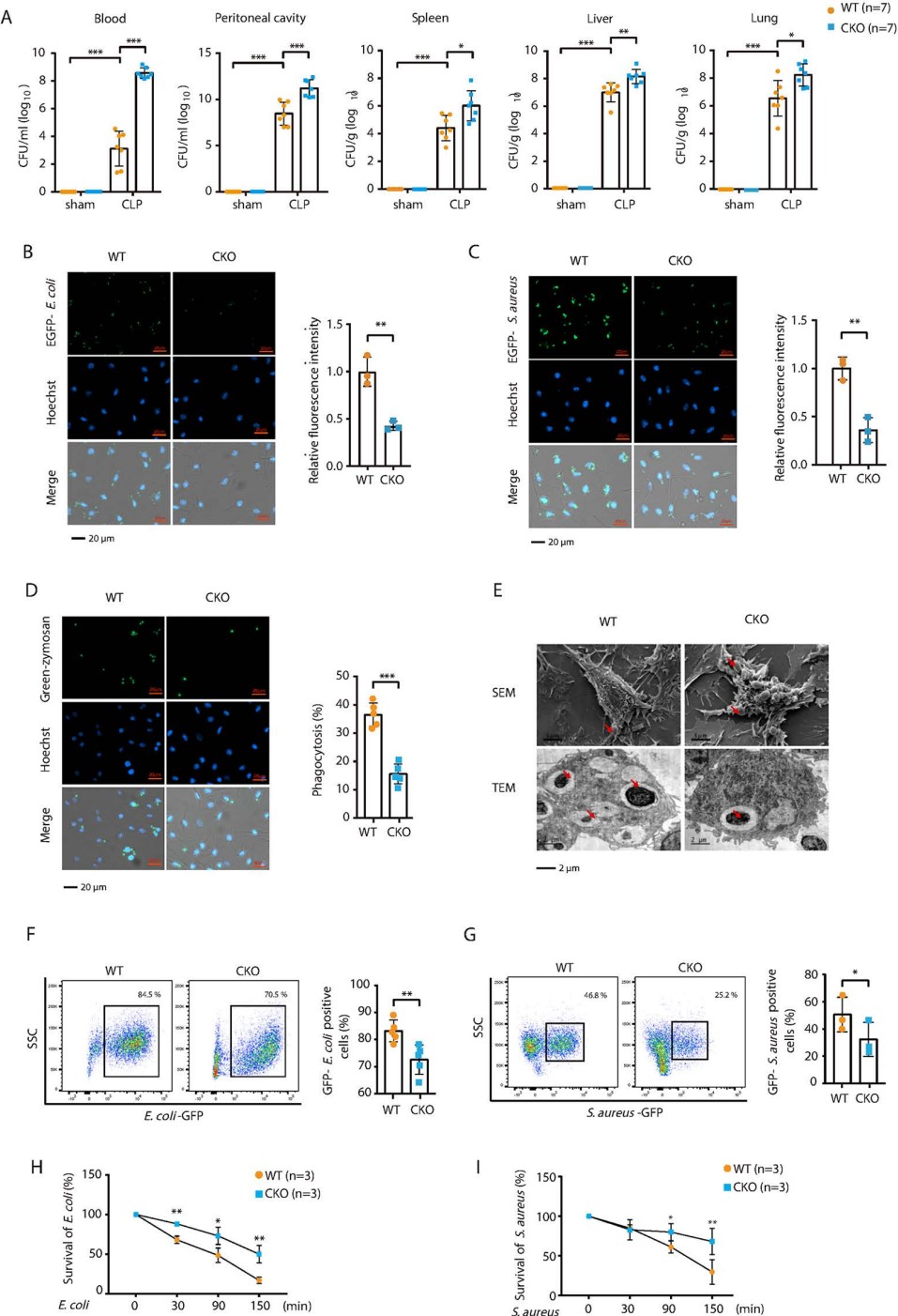

**Fig 3. BRD4 deficiency impairs the phagocytic and bactericidal functions of macrophages. (A)** Bacterial burden in the blood, peritoneal fluid, spleen, liver, and lung of WT and *Brd4*-CKO mice following sham or CLP surgery, assessed 24 hours post-surgery (n = 7). **(B & C)** WT and *Brd4*-deficient BMDMs were infected with GFP-labeled *E. coli* (MOI = 25) (B) or *S. aureus* (MOI = 10) (C) for 1 hours. Bacterial phagocytosis was quantified by fluorescence microscopy (n = 3). **(D)** WT and *Brd4*-deficient BMDMs were challenged with Green-Zymosan, and phagocytosis was quantified by fluorescence microscopy (n = 5). **(E)** Microscopy analysis of zymosan phagocytosis in WT and *Brd4*-deficient BMDMs. Cells were incubated with zymosan (20 particles/cell) for 30 minutes at 37°C, and phagocytosis was analyzed using scanning electron microscopy (SEM) and transmission electron microscopy (TEM). **(F & G)** GFP-labeled *E. coli* (1.0 × 10⁷ CFU) or *S. aureus* (1.0 × 10⁷ CFU) were injected into the peritoneal cavities of WT or *Brd4*-CKO mice. After 15 minutes, mice were sacrificed, and peritoneal macrophages were analyzed for phagocytic activity by flow cytometry. **(H&I)** WT and *Brd4*-deficient

BMDMs were infected with *E. coli* (MOI = 50) (H) or *S. aureus* (MOI = 10) **(I)**, and bacterial killing function was assessed (n = 3). Data are expressed as means ± SD. Statistical analyses were performed using two-way ANOVA with Tukey's post-hoc test for multiple comparisons. *p < 0.05, **p < 0.01, ***p < 0.001.

*E. coli* or *S. aureus*. Compared to *S. aureus*, *E. coli* induced more significant transcriptomic changes in both WT and *Brd4*-deficient BMDMs (Fig 4A, 4B). Importantly, we observed substantial differences in gene expression between WT and *Brd4*-deficient BMDMs upon exposure to either bacterium (S4A and S4B Fig). These differences were primarily associated with immune-related genes involved in phagosome formation, antigen processing and presentation, the B cell receptor signaling pathway, and the TNF signaling pathway (Fig 4C, 4D). Notably, we identified distinct expression patterns for genes encoding key phagocytic receptors, including two class A scavenger receptors: MARCO and MSR1. Both receptors were significantly downregulated in *Brd4*-deficient BMDMs (Fig 4E, 4F).

To confirm these findings, we performed quantitative PCR (qPCR) on WT and *Brd4*-deficient BMDMs infected with either *E. coli* or *S. aureus*, which further validated the reduced expression of *Marco* and *Msr1* (Fig 4G, 4H). Western blotting and immunofluorescence assays revealed that bacterial infection led to upregulation of MARCO protein expression in WT BMDMs; however, this upregulation was absent in *Brd4*-deficient BMDMs (Figs 4I-K and S4C). Furthermore, we observed that MARCO protein levels were lower in peritoneal macrophages from CLP-induced WT mice compared to sham-treated WT mice, with an even further reduction in CLP-induced *Brd4*-CKO mice (Fig 4L). These findings suggest that BRD4 plays a crucial role in regulating MARCO and MSR1 expression in macrophages.

## BRD4 regulates *Marco* and *Msr1* transcription through NRF2 activation

To investigate the molecular mechanism by which BRD4 regulates the expression of MARCO and MSR1, we analyzed publicly available Chromatin Immunoprecipitation sequencing (ChIP-seq) data. Although no data related to *E. coli* or *S. aureus* infection in macrophages were available, we identified LPS treatment data, as LPS is known to induce the transcription of both *Marco* and *Msr1* [29] (S5A Fig). Our analysis revealed that BRD4 was significantly recruited to the promoter regions of both *Marco* and *Msr1* following LPS stimulation. Notably, each of these regions contained two NRF2 binding sites (Figs 5A and S5B), suggesting that BRD4 may play a pivotal role in the regulation of these genes through NRF2-mediated mechanisms. This hypothesis was supported by ChIP experiments, which demonstrated that, during *E. coli* or *S. aureus* infection, the binding of BRD4 and NRF2 to the *Marco* gene promoter region was significantly increased. Furthermore, BRD4 depletion led to a notable reduction in NRF2 binding (Fig 5B, 5C). Immunoprecipitation assays revealed enhanced endogenous BRD4-NRF2 interaction following bacterial infection (Fig 5D, 5E), and APEX2 proximity labeling confirmed their predominant co-localization in the nucleus (S5C Fig). These data indicate that bacterial infection promotes the formation of a nuclear BRD4-NRF2 complex that enhances *Marco* and *Msr1* transcription.

Further analysis indicated that both *E. coli* and *S. aureus* infections led to an upregulation of NRF2 expression. However, in *Brd4*-deficient BMDMs, NRF2 protein levels were significantly reduced, while mRNA levels remained unchanged (Figs 5F, 5G, S5D). This suggests that BRD4 primarily regulates NRF2 at the protein level. To test this hypothesis, we treated bacteria-infected cells with the protein synthesis inhibitor cycloheximide (CHX). The half-life of NRF2 was approximately 25 minutes in WT BMDMs and 15 minutes in *Brd4*-deficient BMDMs (Fig 5H, 5I), indicating that BRD4 is essential for maintaining NRF2 stability. Ubiquitination and subsequent proteasome-mediated degradation are key mechanisms regulating intracellular NRF2 levels [39]. To investigate the impact of BRD4 on NRF2 ubiquitination, we co-transfected WT or *Brd4* knockout 293T cells with Flag-NRF2 and HA-Ubiquitin. Notably, *Brd4* knockout resulted in a significant increase in NRF2 ubiquitination (Fig 5J). Consistent with these findings, *S. aureus* infection increased endogenous NRF2 expression and decreased NRF2 ubiquitination in WT BMDMs, but in *Brd4*-deficient BMDMs, NRF2 expression was reduced and

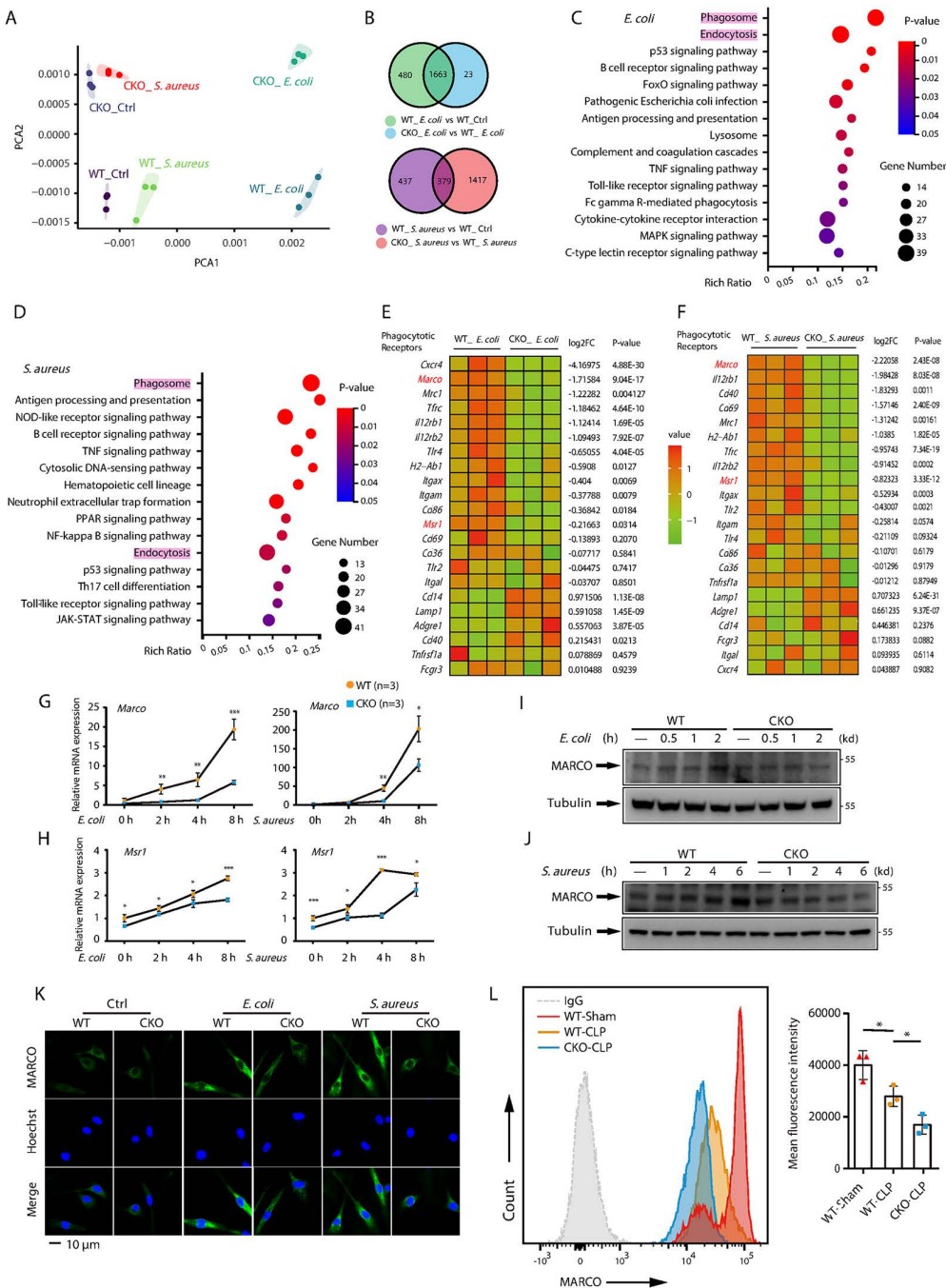

**Fig 4. Macrophages from *Brd4*-CKO mice exhibit reduced expression of MARCO and MSR1. (A)** Principal component analysis (PCA) of RNA-Seq data from WT and *Brd4*-deficient BMDMs challenged with or without *E. coli* or *S. aureus*. **(B)** Venn diagrams depicting the number of genes with significantly altered expression (fold change ≥ 2; adjusted P value ≤ 0.05) in untreated or *E. coli*-challenged WT and *Brd4*-deficient BMDMs (upper) and in untreated or *S. aureus*-challenged WT and *Brd4*-deficient BMDMs (lower). **(C & D)** Gene ontology (GO) analysis of significantly altered gene clusters in *E. coli* (C) or *S. aureus* (D) challenged WT and *Brd4*-deficient BMDMs. **(E & F)** Heat maps illustrating the relative expression levels of phagocytic receptor genes in *E. coli* (E) or *S. aureus* (F) challenged WT and *Brd4*-deficient BMDMs. **(G & H)** WT and *Brd4*-deficient BMDMs were challenged with or without *E. coli* (G) or *S. aureus* (H) for the indicated times. Marco (G) and Msr1 (H) mRNA levels were quantified by qRT-PCR. **(I & J)** WT and *Brd4*-deficient BMDMs were challenged with or without *E. coli* (I) or *S. aureus* (J) for the indicated times. MARCO protein levels were analyzed by immunoblotting. **(K)** Immunofluorescence staining of MARCO in WT and *Brd4*-deficient BMDMs after infection with or without *E. coli* or *S. aureus* for 6 hours.

(L) MARCO protein expression in peritoneal macrophages from WT and *Brd4*-CKO mice (n = 3) was measured 24 hours after sham or CLP surgery by flow cytometry. Data are expressed as means ± SD. Statistical analyses were performed using two-way ANOVA with Tukey's post-hoc test for multiple comparisons. *p < 0.05, **p < 0.01, ***p < 0.001.

ubiquitination was elevated (Fig 5K). These data indicate that BRD4 stabilizes NRF2 protein by inhibiting ubiquitin-proteasome-mediated degradation.

Translocation of NRF2 into the nucleus is a critical step for the regulation of downstream gene expression [40]. We next investigated whether BRD4 modulates the intracellular distribution of NRF2. Immunofluorescence assays showed that *E. coli* and *S. aureus* infections promoted NRF2 accumulation in the nucleus, but BRD4 deficiency led to reduced NRF2 accumulation in the nucleus (Fig 5L). Nuclear-cytoplasmic fractionation experiments further confirmed that BRD4 deficiency diminished the accumulation of NRF2 in the nucleus following *S. aureus* infection (Fig 5M). To further confirm that BRD4 regulates the transcriptional activity of NRF2, we measured the mRNA levels of well-established NRF2 target genes. In addition to *Marco* and *Msr1*, expression of other Nrf2 target genes, such as *Nqo1* and *Gclc*, was significantly reduced in *Brd4*-deficient BMDMs (S5E and S5F Fig), providing additional evidence for the impact of BRD4 on NRF2 activity. Taken together, these results demonstrate that BRD4 enhances the transcription of *Marco* and *Msr1* by stabilizing NRF2 protein and facilitating its nuclear translocation.

## BRD4 interacts with the DLG and ETGE motifs of NRF2 through its C-terminal domain

To further investigate the molecular mechanism by which BRD4 activates NRF2, we conducted a detailed analysis of their interaction. Immunoprecipitation assays following overexpression of both BRD4 and NRF2 confirmed their direct interaction (Fig 6A, 6B). NRF2 contains six distinct domains (Neh1–6) (Fig 6C), and we identified the N-terminal Neh2 domain as essential for its interaction with BRD4 (Fig 6D, 6E). The DLG and ETGE motifs within the Neh2 domain are crucial for NRF2 binding to KEAP1 [41,42]. Notably, when the DLG and ETGE motifs were deleted from NRF2, the interaction with BRD4 was abolished (Fig 6F, 6G), suggesting that BRD4 likely competes with KEAP1 for binding to NRF2.

To identify the specific domain(s) of BRD4 responsible for mediating this interaction, we constructed a series of BRD4 mutants with deletions of individual domains (Fig 6H). Interestingly, in contrast to many previously characterized BRD4-binding partners, the BRD4-NRF2 interaction did not depend on the BRD4 bromodomains (BDs), as deletion of both BD1 and BD2 had no impact on the BRD4-NRF2 interaction (Fig 6I). Further analysis revealed that the C-terminal domain (CTD) of BRD4 is critical for binding NRF2 (Fig 6J, 6K). Together, these results demonstrate that BRD4 interacts with NRF2 through its CTD, specifically recognizing the DLG and ETGE motifs within the Neh2 domain of NRF2.

## BRD4 enhances NRF2 stability by inhibiting the KEAP1-NRF2 interaction

NRF2's DLG and ETGE motifs are known to interact with the Kelch domain of KEAP1, with the three consecutive arginine/lysine residues (R/K) within the Kelch domain playing a crucial role in this interaction [43,44]. While the C-terminal domain (CTD) of BRD4 lacks a sequence similar to the KEAP1 Kelch domain, our analysis revealed several key motifs within the BRD4 CTD, including RRR, RKR, and RRRR (Fig 7A). To explore the importance of these motifs in the BRD4-NRF2 interaction, we generated a series of deletion mutants of BRD4 (1–1279, 1–1328, and 1–1334) and assessed their binding affinity to NRF2. Compared to WT BRD4 (1–1362), all deletion mutants displayed significantly reduced binding to NRF2 (Fig 7B). Notably, the 1–1279 and 1–1328 mutants, which lacked the RKR and RRRR motifs, exhibited comparable reductions in binding, even though the 1–1328 mutant retained the RRR motif. Conversely, the 1–1334 mutant, which retained the RKR motif but lacked the RRRR motif, showed an enhanced ability to bind NRF2. These findings suggest that the C-terminal RKR and RRRR motifs are crucial for the BRD4-NRF2 interaction, while the RRR motif alone is not essential.

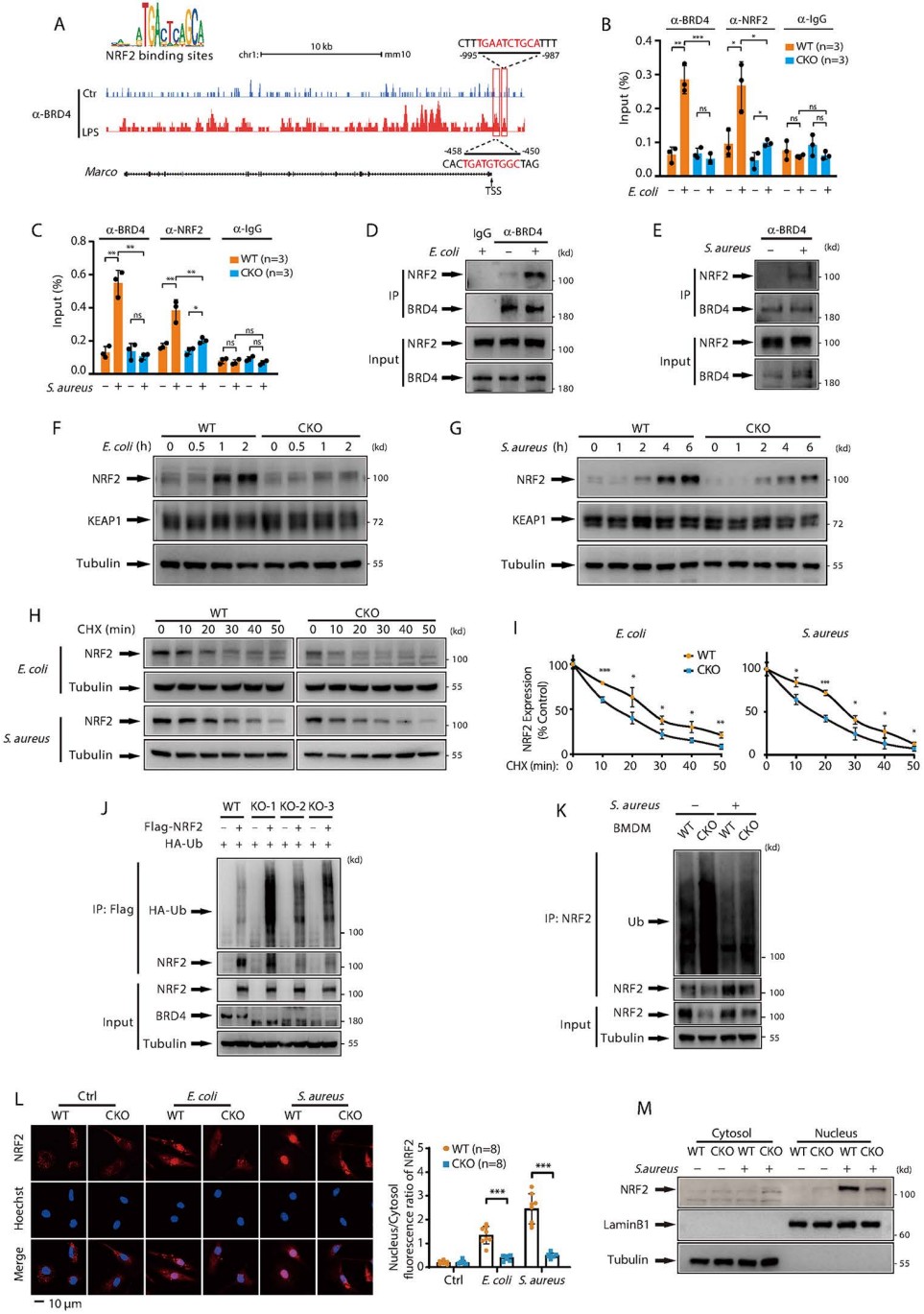

**Fig 5. BRD4 regulates *Marco* and *Msr1* transcription through NRF2 activation. (A)** Schematic representation of the potential NRF2 binding sites within the murine *Marco* promoter region, as identified by BRD4 chromatin immunoprecipitation (ChIP)-sequencing. The diagram was adapted from the GEO Datasets of the National Center for Biotechnology Information (GSE113226) (https://www.ncbi.nlm.nih.gov/geo/query/acc.cgi?acc=GSE113226). TSS: transcription start site. **(B & C)** WT and *Brd4*-deficient BMDMs were infected with or without *E. coli* (B) or *S. aureus* (C) for 4 hours. ChIP assays were performed using antibodies against BRD4, NRF2 and IgG, and the *Marco* promoter was analyzed by qPCR. **(D & E)** WT BMDMs were infected with or without *E. coli* (D) or *S. aureus* (E) for 4 hours. Endogenous BRD4 was immunoprecipitated, and associated NRF2 was detected by immunoblotting. IgG served as a control. **(F & G)** WT and *Brd4*-deficient BMDMs were infected with or without *E. coli* (F) or *S. aureus* (G) for the indicated times. NRF2 and KEAP1 protein levels were analyzed by immunoblotting. **(H)** WT and *Brd4*-deficient BMDMs were infected with *E. coli* (upper) or *S. aureus* (bottom) for 4 hours, followed by cycloheximide (CHX) (5 µg/mL) treatment for the indicated times. NRF2 protein levels were analyzed by

immunoblotting. **(I)** Quantification of NRF2 protein levels from panel H is shown (n = 3). **(J)** WT and *Brd4*-knockout HEK293T cells were co-transfected with Flag-NRF2 and HA-ubiquitin (Ub) plasmids for 48 hours and treated with MG132 (10 µg/mL) for 4 hours before being harvested. Exogenous NRF2 was immunoprecipitated, and the ubiquitination of NRF2 was determined by immunoblotting. **(K)** WT and *Brd4*-deficient BMDMs were infected with or without *S. aureus* for 4 hours. Endogenous NRF2 was immunoprecipitated, and the ubiquitination of NRF2 was determined by immunoblotting. **(L)** Immunofluorescence staining of NRF2 in WT and *Brd4*-deficient BMDMs after *E. coli* or *S. aureus* infection for 6 hours (left), and the relative quantification of NRF2 nuclear-cytoplasmic distribution (right). **(M)** Nuclear-cytoplasmic fractionation of NRF2 in WT and *Brd4*-deficient BMDMs after *S. aureus* infection for 6 hours. Data are expressed as means ± SD. Statistical analyses were performed using two-way ANOVA with Tukey's post-hoc test for multiple comparisons. *p < 0.05, **p < 0.01, ***p < 0.001. ns, not significant.

To further investigate the functional importance of the RKR and RRRR motifs in BRD4, we introduced mutations into these sequences, substituting methionine residues to generate the 3M-BRD4, 4M-BRD4, and 7M-BRD4 mutants (Fig 7A). When compared to WT BRD4, both the 3M-BRD4 and 4M-BRD4 mutants exhibited a reduction in their ability to bind NRF2. Importantly, the 7M-BRD4 mutant demonstrated a near-complete loss of interaction with NRF2 (Fig 7C), highlighting the critical role of both motifs in mediating the BRD4-NRF2 interaction. Moreover, sequence conservation analysis revealed that the RKR and RRRR motifs are highly conserved across species (Fig 7D), suggesting an evolutionarily conserved interaction mechanism.

Given the established interaction between BRD4 and NRF2's DLG and ETGE motifs, we hypothesized that BRD4 may compete with KEAP1 for binding to NRF2. Indeed, when WT BRD4 was overexpressed, the interaction between NRF2 and KEAP1 was significantly reduced, whereas the 7M-BRD4 mutant had no such effect (Fig 7E, 7F). To directly assess this competition *in vitro*, we generated a GST-tagged BRD4 deletion mutant (BRD4-PID, amino acids 1302–1362), encompassing the NRF2-binding region, as GST-tagged WT BRD4 was unavailable. BRD4-PID bound to NRF2 and recapitulated the inhibitory effect of WT BRD4 on the NRF2–KEAP1 interaction in co-immunoprecipitation assays (Fig 7G). Moreover, increasing amounts of GST-BRD4-PID progressively diminished recombinant NRF2 binding to KEAP1 (Fig 7H). Consistent with these findings, WT BRD4, but not 7M-BRD4, led to a reduction in the ubiquitination of NRF2 by KEAP1 (Fig 7I). To assess the effect on NRF2 stability, we evaluated NRF2 levels in *BRD4* knockout 293T cells overexpressing either WT or 7M BRD4. Consistently, NRF2 was more stable in cells overexpressing WT BRD4 compared to those expressing 7M BRD4 (Fig 7J, 7K). These findings suggest that BRD4 enhances NRF2 stability by disrupting the interaction between NRF2 and KEAP1, thereby reducing NRF2 ubiquitination and degradation.

### Myeloid-specific Nrf2 expression rescues the susceptibility of *Brd4*-CKO mice to CLP-induced sepsis

Although our *in vitro* data indicated that BRD4 enhances macrophage antibacterial immunity by stabilizing and activating NRF2, whether NRF2 restoration *in vivo* could rescue the defective antimicrobial function in *Brd4*-CKO septic mice remained unclear. To address this, we utilized an adeno-associated virus type 9 (AAV9) system to achieve myeloid-specific expression of NRF2 under control of the Lyz2 promoter in *Brd4*-CKO mice. Successful NRF2 restoration was confirmed 21 days post-infection (Fig 8A). Immunostaining showed that AAV9-Lyz2-*Nrf2* infection restored NRF2 levels in BMDMs from *Brd4*-CKO mice and consequently increased expression of the downstream targets MARCO and MSR1 (Fig 8A). We then subjected WT and *Brd4*-CKO mice, with or without NRF2 restoration, to CLP-induced sepsis (Fig 8B). At 24 hours post-CLP, *Brd4*-CKO mice with restored NRF2 expression exhibited significantly reduced plasma inflammatory cytokine levels and attenuated liver and lung injury compared to *Brd4*-CKO mice without NRF2 restoration (Fig 8B-D). Consistently, bacterial loads in the blood, peritoneal lavage fluid, and various organs were also significantly reduced following NRF2 restoration (Fig 8E). These results indicate that the increased susceptibility of *Brd4*-CKO mice to CLP-induced sepsis is primarily attributable to NRF2 deficiency in myeloid cells.

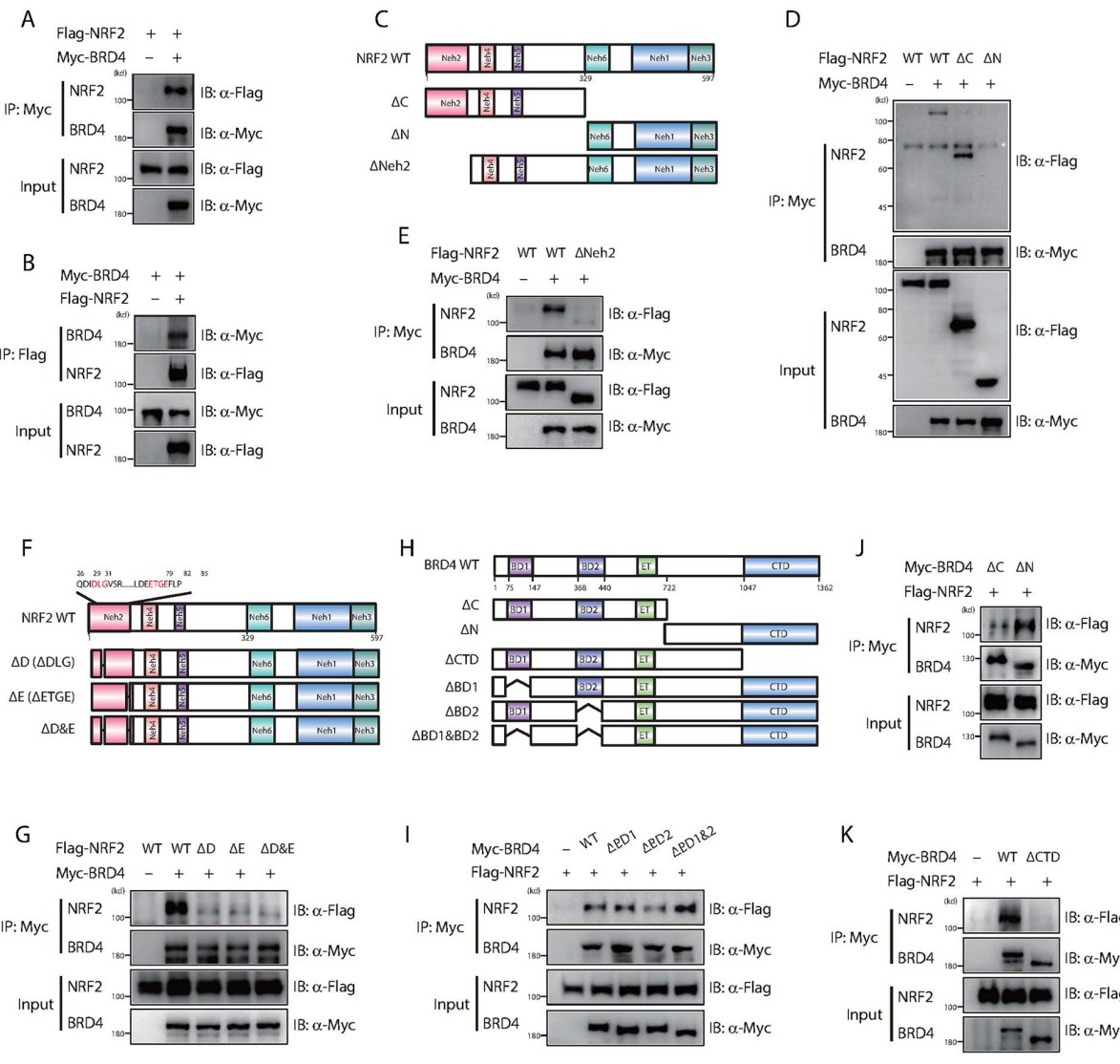

**Fig 6. BRD4 interacts with the DLG and ETGE motifs of NRF2 through its C-terminal domain. (A & B)** HEK293T cells co-expressing Myc-BRD4 and Flag-NRF2 were immunoprecipitated using anti-Myc (A) or anti-Flag (B) antibodies. The interaction was detected by immunoblotting. **(C)** Schematic representation of the domain structure of NRF2 and various NRF2 deletion mutants. **(D & E)** HEK293T cells were co-transfected with Myc-BRD4 and Flag-NRF2 or its mutants as indicated in panel **C**. Cell lysates were immunoprecipitated with anti-Myc beads and detected by immunoblotting. Non-specific protein bands are marked with asterisks. **(F)** Schematic representation of the domain structure of NRF2 and its various deletion mutants, with the deleted motifs located within the Neh2 domain of NRF2. **(G)** HEK293T cells were co-transfected with Myc-BRD4 and Flag-NRF2 or its mutants, as indicated in panel **F**. Cell lysates were immunoprecipitated with anti-Myc beads and detected by immunoblotting. **(H)** Schematic diagram of the domain structure of BRD4 and various BRD4 deletion mutants. **(I-K)** HEK293T cells were co-transfected with Flag-NRF2 and Myc-BRD4 or its mutants, as indicated in panel **H**. Cell lysates were immunoprecipitated with anti-Myc beads and detected by immunoblotting.

## The NRF2 agonist SFN improves survival and reduces bacterial burden in both WT and *Brd4*-CKO septic mice

Our findings indicate that the impaired antimicrobial function observed in *Brd4*-CKO mice is linked to reduced NRF2 activation. Consequently, we explored whether pharmacological activation of NRF2 could restore their antimicrobial capacity. Sulforaphane (SFN), a natural isothiocyanate found in cruciferous vegetables, is a well-known pharmacological activator of NRF2 that promotes its release from KEAP1 [45,46]. We first assessed the effects of SFN on the expression of NRF2,

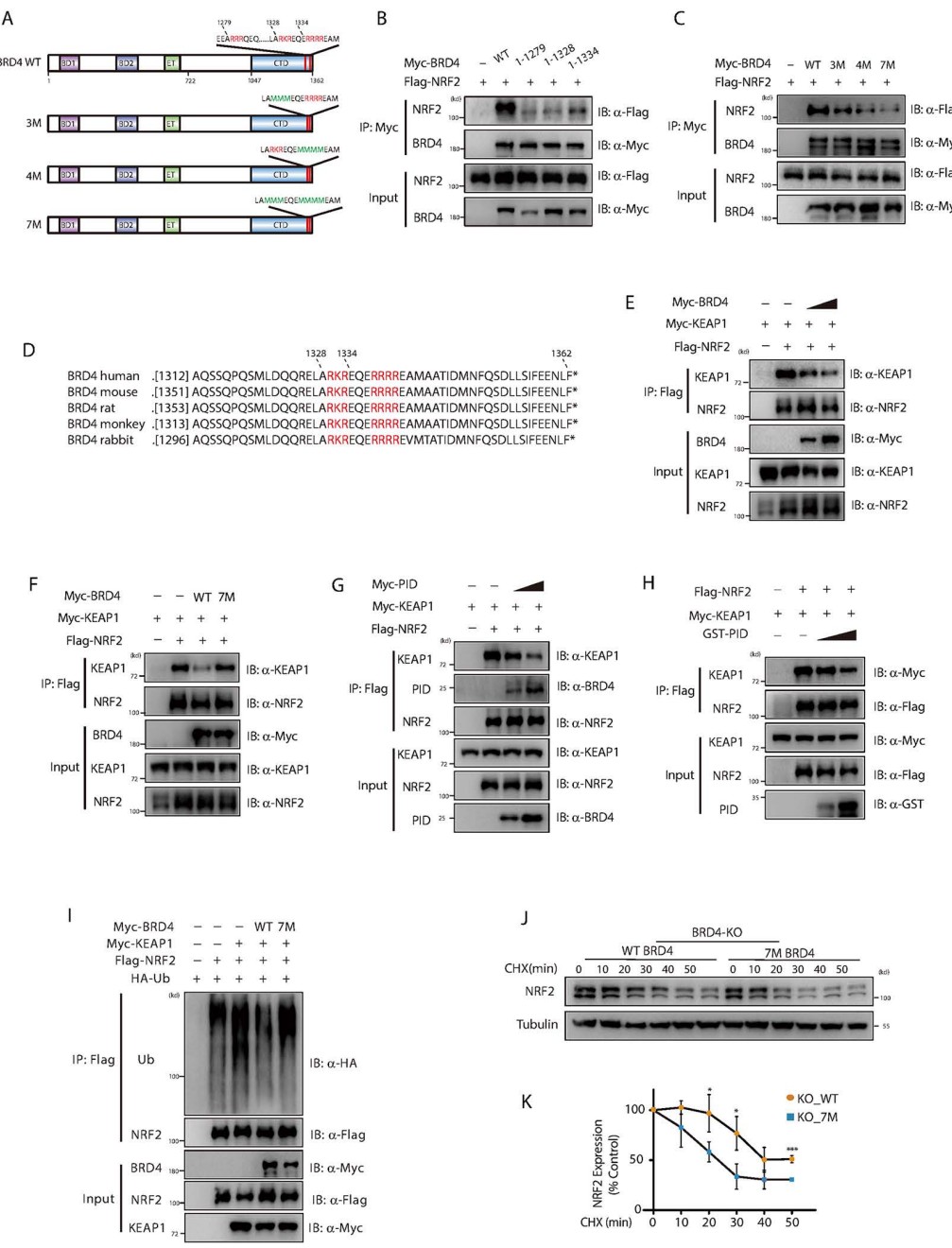

**Fig 7. BRD4 enhances NRF2 stability by inhibiting the KEAP1-NRF2 interaction. (A)** Schematic representation of the BRD4 domain structure, with the C-terminal domain (CTD) containing potential NRF2-binding motifs. **(B)** HEK293T cells were co-transfected with Flag-NRF2 and Myc-BRD4 or its deletion mutants, as indicated. Cell lysates were immunoprecipitated with anti-Myc beads and analyzed by immunoblotting. **(C)** HEK293T cells were co-transfected with Flag-NRF2 and Myc-BRD4 or its point mutants, as indicated in panel **A.** Cell lysates were immunoprecipitated with anti-Myc beads and analyzed by immunoblotting. **(D)** Sequence alignment of BRD4 showing potential NRF2-binding motifs across different species. **(E)** HEK293T cells were co-transfected with Flag-NRF2, Myc-KEAP1, and increasing amounts of Myc-BRD4 plasmids. Cell lysates were immunoprecipitated with anti-Flag beads and analyzed by immunoblotting. **(F)** HEK293T cells were co-transfected with Flag-NRF2, Myc-KEAP1, and Myc-BRD4 or its mutants, as indicated. Cell lysates were immunoprecipitated with anti-Flag beads and analyzed by immunoblotting. **(G)** HEK293T cells were co-transfected with Flag-NRF2, Myc-KEAP1, and Myc-BRD4 PID, as indicated. Cell lysates were immunoprecipitated with anti-Flag beads and analyzed by immunoblotting. **(H)** Recombinant Flag-NRF2, Myc-KEAP1, and increasing amounts of GST-PID were incubated, immunoprecipitated using anti-Flag beads, and analyzed by immunoblotting. **(I)** HEK293T cells were co-transfected with Flag-NRF2, Myc-BRD4, HA-ubiquitin (Ub), and Myc-BRD4 or its mutants for 48

hours, followed by treatment with MG132 (10 µg/mL) for 4 hours. Exogenous NRF2 was immunoprecipitated, and NRF2 ubiquitination was analyzed by immunoblotting. **(J)** BRD4 knockout HEK293T cells were transfected with Myc-BRD4 or its mutants for 48 hours, followed by treatment with cycloheximide (CHX, 5 µg/mL) for the indicated time points. NRF2 protein levels were assessed by immunoblotting. **(K)** Quantification of NRF2 protein levels from (J) were shown (n = 3). Data are expressed as means ± SD. Statistical analyses were performed using two-way ANOVA with Tukey's post-hoc test for multiple comparisons. *p < 0.05, ***p < 0.001.

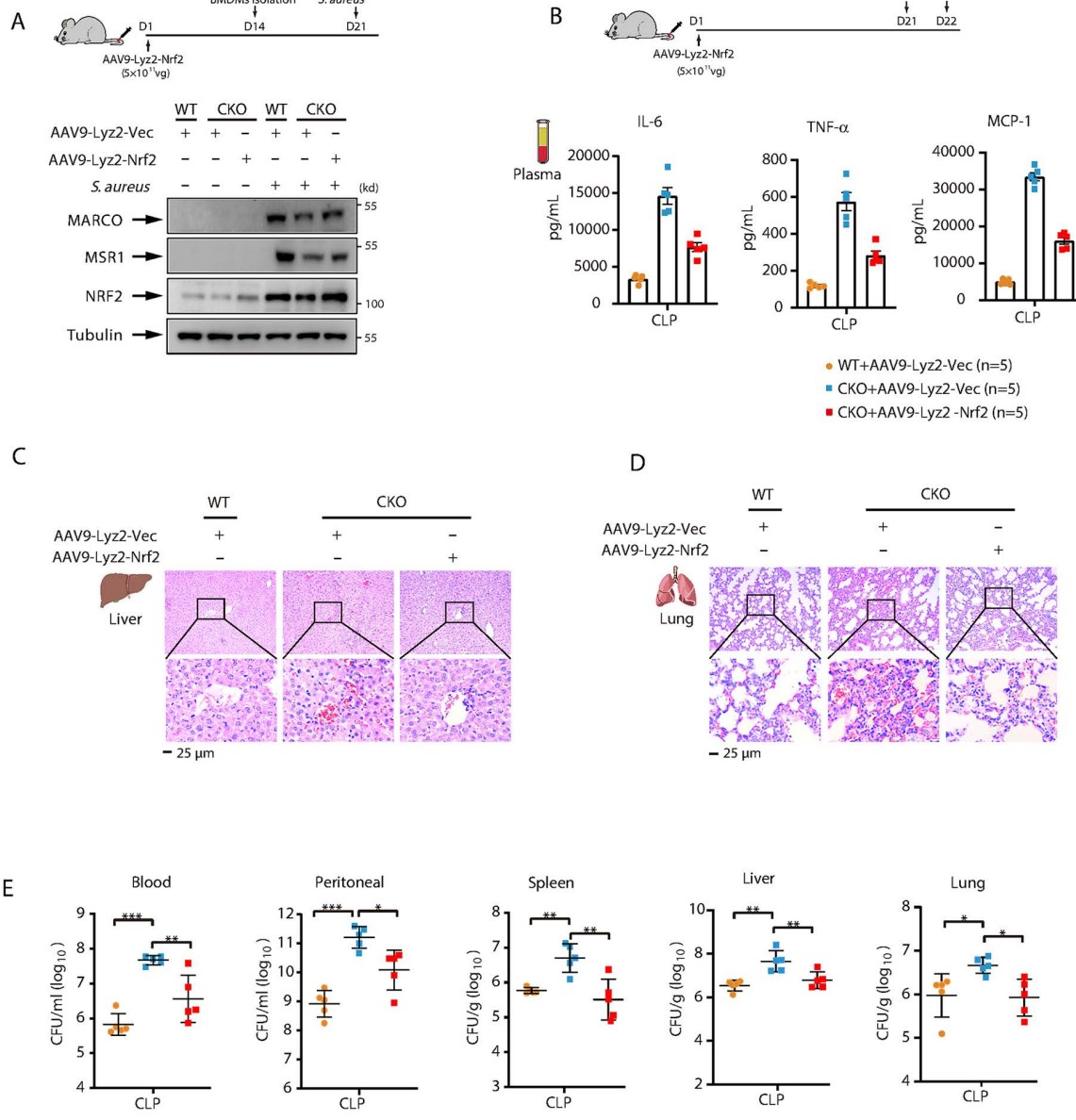

**Fig 8. Myeloid-specific NRF2 restoration rescues antimicrobial function and attenuates sepsis severity in *Brd4*-CKO mice.** WT and *Brd4*-CKO mice were intravenously administered AAV9-Lyz2-Vector or AAV9-Lyz2-NRF2 (5 x 10¹¹ vg) for three weeks. for three weeks prior to CLP surgery. **(A)** Immunoblot analysis of NRF2, MARCO, and MSR1 in BMDMs from the indicated mice following S. aureus infection (MOI = 10, 4 **h**). **(B)** Plasma levels of IL-6, TNF-α, and MCP-1 at 24 h post-CLP (n = 5). **(C and D)** Representative H&E-stained sections of liver (C) and lung (D) from the indicated mice at 24 h post-CLP. **(E)** CFU in blood, peritoneal lavage fluid, lung, spleen, and liver at 24 h post-CLP (n = 5). Data are expressed as means ± SD. Statistical analyses were performed using one-way ANOVA with Tukey's post-hoc test for multiple comparisons. *p < 0.05, **p < 0.01, ***p < 0.001. ns, not significant.

MARCO, and MSR1 in both WT and *Brd4*-deficient BMDMs. Compared to WT BMDMs, *Brd4*-deficient BMDMs exhibited significantly reduced protein levels of NRF2, MARCO, and MSR1 following infection (Fig 9A). Notably, SFN treatment significantly restored the expression of all three proteins in *Brd4*-deficient BMDMs, while co-treatment with ML385, a selective NRF2 inhibitor, reversed these effects in a dose-dependent manner, confirming the specificity of SFN for NRF2 (Figs 9A and S6A). SFN treatment also upregulated NRF2, MARCO, and MSR1 expression in WT BMDMs as well (Fig 9A), suggesting that SFN enhances NRF2 activation and promotes the expression of scavenger receptors in both WT and *Brd4*-deficient macrophages.

To assess the functional consequences of SFN treatment, we evaluated the phagocytic capabilities of WT and *Brd4*-deficient BMDMs against both *S. aureus* and *E. coli*. SFN treatment significantly enhanced the phagocytic capacity of both WT and *Brd4*-deficient BMDMs, as shown by the increased engulfment of these bacterial pathogens (Figs 9B and S6B). Consistent with its effects on protein expression, ML385 abolished SFN-mediated enhancement of phagocytosis (S6C Fig). These results demonstrate that SFN restores antimicrobial function in *Brd4*-deficient macrophages through NRF2-dependent upregulation of scavenger receptors.

We next examined the effects of SFN *in vivo* using WT and *Brd4*-CKO septic mice. To assess the potential protective effects of SFN, we employed a more severe sepsis model by increasing the ligation length or needle thickness during CLP, as described in previous studies [47,48]. In this model, survival rates of WT and *Brd4*-CKO septic mice were 25% and 0%, respectively, at day 8 post-CLP surgery. However, following SFN treatment, survival rates increased to approximately 75% for WT and 50% for *Brd4*-CKO septic mice (Fig 9C). These findings suggest that SFN treatment improves survival in both WT and *Brd4*-CKO septic mice. To confirm the protective effects of SFN, we assessed organ injury in septic mice. Histological examination revealed significantly reduced damage in the livers, lungs, and spleens of SFN-treated mice (Fig 9D-F). We also examined the bacterial load in various tissues during CLP-induced sepsis. SFN treatment significantly reduced bacterial burden in the peritoneal cavity, liver, lungs, and spleen (Fig 9G). these results demonstrate that SFN decreases mortality, bacterial load, and organ injury in both WT and *Brd4*-CKO septic mice.

To confirm that the protective effects of SFN in *Brd4*-CKO septic mice are mediated through NRF2 activation, we administered ML385 to SFN-treated *Brd4*-CKO mice subjected to CLP. ML385 treatment reversed the SFN-mediated reduction in plasma inflammatory cytokine levels, attenuated organ injury, and increased bacterial clearance, demonstrating that SFN acts specifically through NRF2 in this context (S7A–C Fig). Collectively, these results demonstrate that SFN reduces mortality, bacterial load, and organ injury in septic mice through NRF2-dependent mechanisms.

Finally, we investigated the association between NRF2 expression levels and the sepsis severity in clinical cohorts. *NRF2* expression in monocytes from PBMCs exhibited a significant inverse correlation with SOFA scores (Fig 9H), suggesting that lower NRF2 expression may be associated with more severe sepsis. Additionally, *NRF2* expression positively correlated with *MARCO* expression in the whole blood of septic patients (Fig 9I). Notably, *MARCO* expression was significantly lower in non-survivors compared to survivors (Fig 9J), remaining robust after adjusting for age and sex (S1 Table). These findings suggest that NRF2 may represent a potential therapeutic target for sepsis in humans.

## Discussion

Macrophages are critical in defending against pathogens and play a central role in eradicating bacterial infections during sepsis. Impaired macrophage antibacterial function is a hallmark of sepsis pathophysiology [49–51]. Here, we identify the impaired BRD4-NRF2-MARCO/MSR1 pathway as a critical mechanism underlying macrophage dysfunction in sepsis (Fig 10). Notably, septic patients with reduced *BRD4* expression in monocytes/macrophages from peripheral blood, but not in other immune cells, exhibited more severe inflammatory responses and organ damage (Figs 1, 2). These findings highlight BRD4 in monocytes/macrophages as both a potential biomarker for disease severity and a promising therapeutic target. However, the precise mechanisms underlying the reduction of BRD4 expression during sepsis remain unclear. While the decreased BRD4 protein levels observed during late-stage infection can be partially attributed to transcriptional

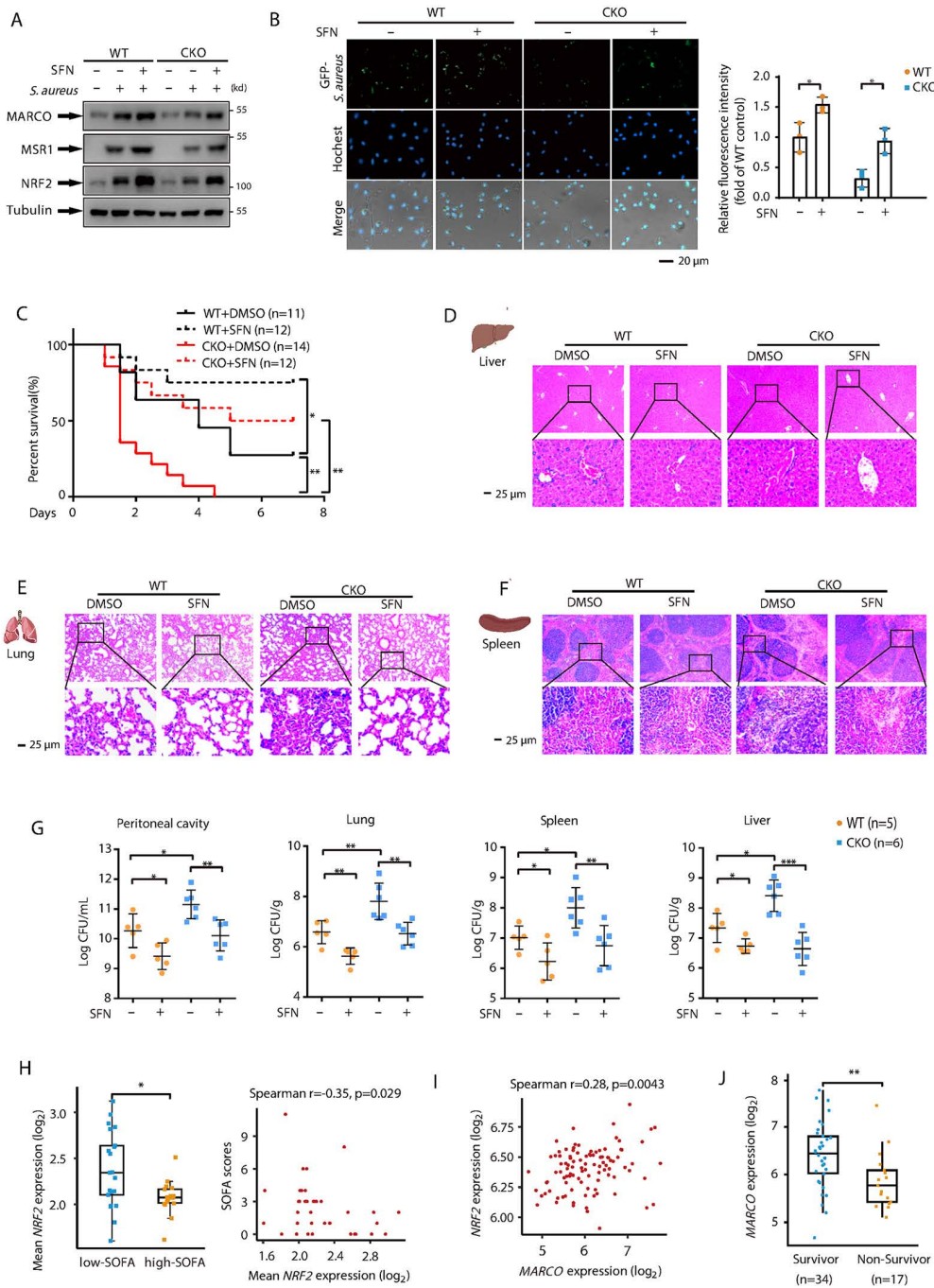

**Fig 9. The NRF2 agonist SFN improves survival and reduces bacterial burden in both WT and *Brd4*-CKO septic mice. (A)** WT and *Brd4*-deficient BMDMs pre-treated with or without the Sulforaphane (SFN, 10 μM) for 1 hour, followed by *S. aureus* treatment for 4 hours. Protein levels of NRF2, MARCO and MSR1 in cell lysates were assessed by immunoblotting. **(B)** WT and *Brd4*-deficient BMDMs pre-treated with or without the SFN (10 μM) for 1 hour, followed by infection with GFP-labeled *S. aureus* for 1 hour. Bacterial phagocytosis was quantified by fluorescence microscopy (n = 3). **(C)** Survival curves of WT and *Brd4*-CKO mice following CLP surgery, with or without intravenous administration of SFN (0.4 mg/kg). Survival was monitored for 7 days (n = 11-14). **(D-F)** Representative H&E-stained images of the liver **(D)**, lung **(E)**, and spleen (F) from WT and *Brd4*-CKO mice 24 hours post-CLP, with or without SFN treatment. **(G)** Bacterial burden in the peritoneal fluids, lung, spleen and liver of mice treated as described in panel D (n = 5–6). **(H)** Comparison of *NRF2* mRNA levels in monocytes between low-SOFA and high-SOFA groups (left). The cutoff for binary classification was set at the median SOFA scores in patients. Scatter plot showing the association between *NRF2* mRNA levels in monocytes (x-axis) and SOFA

scores (y-axis) (right). Data sources: SCP548. **(I)** Correlation between *NRF2* and *MARCO* mRNA levels in whole blood of septic patients. Data sources: GSE95233. **(J)** Comparison of *MARCO* mRNA levels in blood between survivor and non-survivors of septic patients. Data sources: GSE95233. Data presented in panels H–J are expressed as medians ± interquartile range and were analyzed using the Wilcoxon rank-sum test (two-tailed). Other data are shown as means ±SD. Statistical analyses were performed using two-way ANOVA with Tukey's post-hoc test for multiple comparisons. *p < 0.05, **p < 0.01, ***p < 0.001.

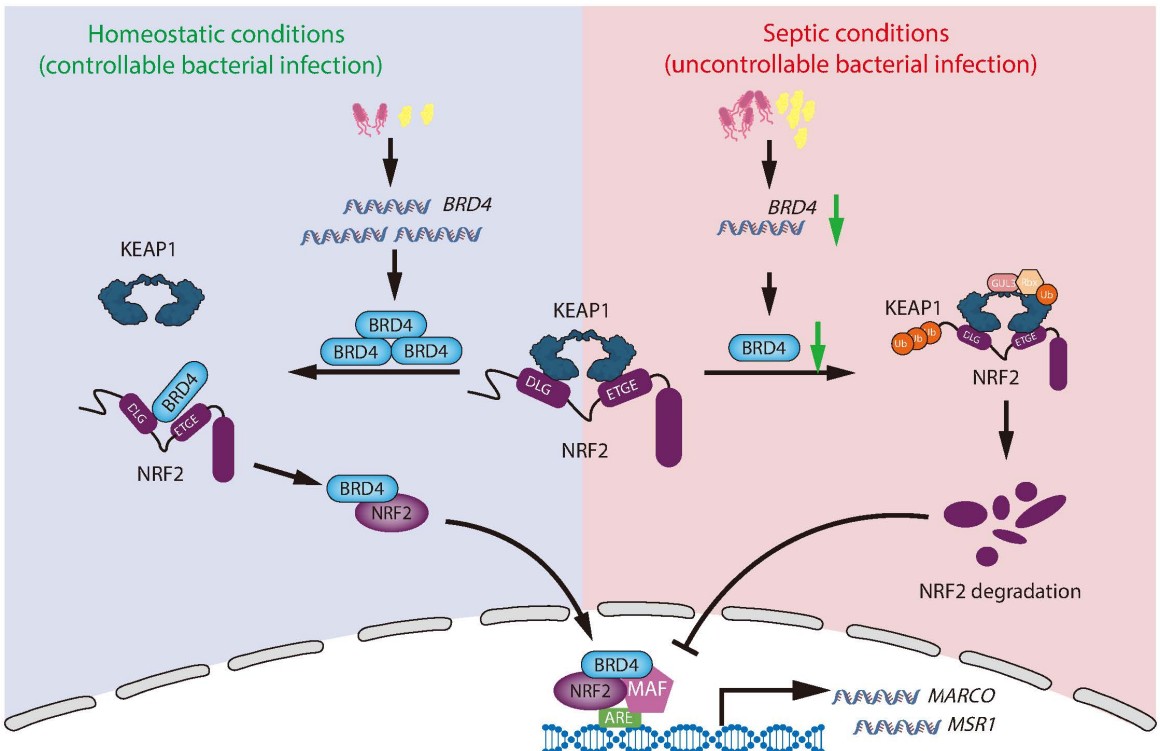

**Fig 10. Schematic model illustrating the role of BRD4 in modulating antimicrobial defense via NRF2 activation in macrophages to confer protection against sepsis.** During the early stages of bacterial infection or in a controlled infection scenario, BRD4 mRNA and protein levels are upregulated. The increased BRD4 competitively binds to KEAP1, thereby reducing the ubiquitination and degradation of NRF2 and promoting its nuclear translocation. This process activates NRF2 target genes, such as MARCO and MSR1, enhancing the antimicrobial capabilities of macrophages. However, during sepsis or uncontrolled bacterial infection, the downregulation of B1RD4 mRNA and protein impairs the BRD4-NRF2-MARCO/MSR1 signaling pathway, resulting in a decline in the antimicrobial function of macrophages.

repression (S1D Fig), post-transcriptional mechanisms, particularly enhanced protein degradation, may also contribute. Further investigation is needed to elucidate the molecular pathways regulating BRD4 expression and the factors driving its downregulation in this context. Unraveling these mechanisms could provide valuable insights into the pathogenesis of sepsis and offer new avenues for targeted therapies aimed at restoring macrophage function.

For our *in vitro* sepsis models, we selected *E. coli* and *S. aureus* as prototypes of Gram-negative and Gram-positive bacteria, respectively, based on established studies [52]. To assess the clinical relevance of these choices, we performed 16S rDNA sequencing on blood from CLP-induced WT mice. The most abundant species detected was *Escherichia*, consistent with the predomination of *E. coli* in CLP-derived sepsis models (S8 Fig). However, we did not detect *Staphylococcus* in our sequencing dataset. This may reflect limited sequencing depth or dynamic shifts in the microbiota during CLP-induced sepsis [53], as *Staphylococcus* species have been reported in mouse cecal contents [54]. Moreover, *S.*

*aureus* remains a significant clinically relevant pathogen in human sepsis [55], warranting its inclusion in our mechanistic studies.

In the *in vitro* bacterial infection model, BRD4 expression in macrophages exhibit a time-dependent dynamic change, with an initial increase followed by a gradual decrease (Fig 1K, 1L). This pattern suggests that BRD4 may be rapidly induced during early infection in response to pathogen stimulation, potentially contributing to the initiation of pro-inflammatory cytokine production [21]. The subsequent downregulation in later stages of infection may reflect a negative feedback mechanism designed to prevent excessive inflammation and tissue damage. Interestingly, we observed a significant reduction in BRD4 expression in monocytes/macrophages from septic patients and mice compared to healthy controls. This seemingly paradoxical finding may be attributed to the fact that the septic samples were predominantly collected during the phase characterized by severe infection and pronounced pathological features, when BRD4 downregulation would be expected. To gain a more comprehensive understanding of BRD4's role in sepsis, future studies should aim to dynamically monitor BRD4 expression in macrophages at different stages of sepsis progression, particularly during the early hyper-inflammatory phase and the later immunosuppressive phase. Such investigations will provide deeper insights into the complex regulatory role of BRD4 in sepsis pathophysiology and could inform the development of stage-specific, targeted therapeutic strategies.

The contrasting phenotypes of *Brd4*-CKO mice in LPS-induced endotoxemia versus CLP-induced polymicrobial sepsis reflect the dual functions of BRD4 in calibrating inflammatory and antimicrobial programs [21]. BRD4 potentiates pro-inflammatory cytokine expression, and its deletion attenuates lethal cytokine storm during LPS challenge, conferring protection. However, effective host defense against live infections requires robust bacterial clearance. BRD4 deficiency impairs phagocytosis by downregulating scavenger receptors MARCO and MSR1, which also induced by LPS and regulated by BRD4 at the promoter level. While our analysis of public ChIP-seq data utilized LPS stimulation, we confirmed BRD4 recruitment to these promoters during bacterial infection (Fig 5B, 5C). Notably, both MARCO and MSR1 knockout mice exhibit resistance to LPS-induced pathology [56–58], further elucidating the context-dependent outcomes. Thus, BRD4 balances inflammation and antimicrobial defense, with the net effect on host survival determined by the nature of the challenge.

*Brd4*-CKO mice were generated using Lyz2-Cre-mediated deletion, resulting in the absence of BRD4 in both macrophages and neutrophils. We assessed the antimicrobial functions of neutrophils in these mice and found that BRD4 deficiency significantly impairs their phagocytic and bactericidal activities (S9 Fig). These findings indicate that BRD4 is crucial not only for macrophages but also for the antimicrobial functions of myeloid innate immune cells, including neutrophils. The protective role of BRD4 in the CLP model stems from its support of both cell types, explaining the heightened susceptibility in *Brd4*-CKO mice. Notably, while BRD4 is essential for neutrophil function, *Brd4* mRNA levels in neutrophils from septic patients do not correlate with systemic inflammatory markers (S1 Fig). This suggests that BRD4 may modulate rapid antimicrobial responses through post-translational modifications rather than solely through expression levels.

The rapid infiltration of myeloid lineage phagocytic leukocytes, predominantly neutrophils and macrophages, into the peritoneal cavity is distinct feature observed in murine models of polymicrobial sepsis, playing a crucial role in shaping the host's innate immune response against microbial infections [59,60]. In response to a septic challenge, a significant influx of neutrophils and macrophages was observed in the peritoneal cavity of WT mice (S10A and S10B Fig). However, the absence of BRD4 resulted in a remarkable decrease in macrophage recruitment, while no apparent impact on neutrophil recruitment was observed (S10A and S10B Fig). These findings highlight the crucial role of BRD4 in facilitating macrophage recruitment to the peritoneum, while its influence on neutrophil recruitment remains to be fully understood. Hence, the reduced recruitment of macrophages to the peritoneal cavity further contributes to the increased susceptibility of *Brd4*-CKO mice to CLP-induced polymicrobial sepsis.

Previous studies have demonstrated that BRD4 is a negative regulator of NRF2 in prostate cancer cells by promoting KEAP1 expression, thereby reducing NRF2 protein stability [61]. However, our findings reveal a distinct mechanism in

macrophages. In contrast to these earlier observations, our data show that BRD4 knockout in macrophages does not affect the mRNA or protein levels of KEAP1 (Figs 5F, 5G, and S11). Furthermore, in *Brd4*-deficient macrophages, the expression of key NRF2 target genes, such as *Nqo1* and *Gclc*, is significantly reduced (S5E and S5F Fig). This downregulation is associated with increased NRF2 ubiquitination, which leads to a corresponding decrease in NRF2 stability and its nuclear translocation (Fig 5F-M). Taken together, our results indicate that BRD4 acts as a positive regulator of NRF2 in macrophages, a role that contrasts with its function in prostate cancer cells. This discrepancy may reflect cell-type- or context-specific differences. These findings not only underscore the complexity of BRD4's regulatory roles across various cell types but also provide novel insights into its nuanced function in immune cells.

To gain a genome-wide perspective on cooperative transcriptional regulation by BRD4 and NRF2, we analyzed publicly available ChIP-seq data. We acknowledge that this bioinformatic approach has inherent limitations, including the absence of anti-NRF2 ChIP-seq data from LPS-treated BMDMs. As an alternative, we analyzed anti-NRF2 ChIP-seq data from LPS+IFNγ-treated KEAP1-deficient BMDMs, in which NRF2 is stabilized and nuclear-localized, partially mimicking the BRD4-mediated NRF2 stabilization we propose. This analysis identified 2,835 DNA regions co-occupied by BRD4 and NRF2 (S12A and S12B Fig). GO enrichment revealed pathways related to immune response activation, regulation of inflammatory response, and response to bacterial molecules (S12C Fig). KEGG pathway analysis showed significant associations with NOD-like receptor signaling, efferocytosis and phagosome (S12D Fig), consistent with known NRF2 and BRD4 functions. Thus, this analysis provides bioinformatic support for a shared transcriptional program linked to innate immune effector functions. Notably, while *Marco* was identified among co-occupied genes, *Msr1* was not. This discrepancy likely reflects context-dependent regulatory mechanisms, possibly arising from differences in cellular stimulation conditions. Future studies using stimulation-matched ChIP-seq will be required to fully validate the direct co-regulation of specific target genes.

An increasing number of proteins have been identified that activate NRF2 by inhibiting its interaction with KEAP1. Most of these proteins function by directly binding to KEAP1, thereby preventing KEAP1-mediated degradation of NRF2 [62,63]. However, KEAP1 also ubiquitinates and degrades a range of substrates beyond NRF2, raising concerns about potential off-target effects when targeting the KEAP1-NRF2 interface [63,64]. In contrast, our study identifies BRD4 as a highly specific, dual-faceted regulator of NRF2, operating through two distinct mechanisms (Fig 5). First, BRD4 directly interacts with NRF2, sterically blocking KEAP1 binding, thereby stabilizing NRF2 and preventing its proteasomal degradation. This complex then translocates to the nucleus, amplifying the NRF2 pool available for transcriptional activity. Second, BRD4 orchestrates the assembly of a transcriptional machinery with NRF2, directing it to the AREs of target genes and enhancing gene activation (Fig 5A–C). Crucially, BRD4's role is modulatory rather than essential: in *Brd4*-deficient macrophages, NRF2 activation alone was sufficient to induce MARCO/MSR1 expression (Fig 9A). This indicates that BRD4 finely orchestrates NRF2-driven transcriptional programs without acting as its master regulator. This dual mechanism not only avoids the potential promiscuity of KEAP1-targeted inhibition but also positions BRD4 as a critical modulator for context-dependent regulation of NRF2 activity, which is central to cellular protection.

In exploring the molecular mechanism by which BRD4 regulates NRF2 ubiquitination, we considered whether BRD4 might influence Cullin3 E3 ligase activity or the KEAP1-CUL3 interaction. Cullin3 facilitates NRF2 ubiquitination through its adaptor protein KEAP1 [65]. However, BRD4 deletion did not affect KEAP1-CUL3 complex formation or the neddylation status of CUL3 (S13A Fig), indicating that BRD4 does not modulate Cullin3 E3 ligase activity in this context. Additionally, BRD4 knockout did not alter the interaction between NRF2 and known deubiquitinases, including USP7, USP11, or DUB3 (S13B–D Fig), excluding the possibility that BRD4 stabilizes NRF2 by recruiting deubiquitinases. These data further support a model in which BRD4 directly competes with KEAP1 for NRF2 binding, thereby protecting NRF2 from ubiquitin-proteasome degradation without affecting the core ubiquitination machinery.

MARCO and MSR1, well-characterized class A scavenger receptors, play a crucial role in bacterial capture and clearance. The absence of either MARCO or MSR1 in mice leads to compromised bacterial control, resulting in elevated

mortality rates [66–68]. Both MARCO and MSR1 are predominantly expressed on macrophages and are upregulated in response to bacterial infections [68,69]. However, in septic mice with *Klebsiella pneumoniae* lung infection, the initial upregulation of MARCO expression is transient and gradually declines over a period of weeks [70]. Moreover, *E. coli* employs virulence factors to activate an inhibitory Fc receptor (FcRc) pathway, or other mechanisms, which counteract MARCO-mediated bacterial phagocytosis, thus exacerbating sepsis [71,72]. These findings suggest that sepsis-causing bacteria may exploit MARCO to evade macrophage-mediated killing, contributing to the persistence of infection. Supporting this hypothesis, a recent study demonstrated that non-surviving septic patients exhibit lower expression levels of MARCO and MSR1 compared to surviving septic patients (Fig 9J) [73]. Therefore, MARCO and MSR1 may be promising candidate targets for sepsis treatment.

Numerous studies have demonstrated that NRF2 activation offers protective effects against a wide range of diseases, such as diabetes, neurodegenerative disorders, and chronic kidney disease [26]. Notably, NRF2 activation has been shown to enhance the antibacterial defense capabilities of macrophages, thereby offering protection against sepsis [73]. Our study suggests that the immune deficiencies in macrophages caused by sepsis, particularly their impaired antibacterial response, are linked to the downregulation of NRF2 due to reduced BRD4 expression (Fig 5). The use of NRF2 activators was found to restore the antibacterial immune function in macrophages with this immune deficiency (Fig 9). Clinically, we observed that the decreased expression of *NRF2* in septic patients may be positively correlated with poor prognosis (Fig 9). These findings underscore the potential of NRF2 activation as a therapeutic strategy to restore immune function and improve outcomes in sepsis. However, it is important to acknowledge the potential risks associated with sustained NRF2 activation. Prolonged activation of NRF2 has been implicated in tumor promotion and may lead to metabolic disturbances [26,74], which could complicate treatment strategies. Therefore, while NRF2 activators hold promise for enhancing immune function, careful consideration of their long-term effects and context-specific applications is essential.

Collectively, our findings highlight a critical role for BRD4 in orchestrating a proper innate immune response during sepsis. Beyond its activation of NRF2, restoring the diminished expression of BRD4 in macrophages using pharmacological activators could represent a novel therapeutic strategy to improve bacterial clearance efficiency in septic conditions. However, given BRD4's involvement in the regulation of inflammatory gene expression, its early-stage activation during sepsis may trigger a severe inflammatory storm. To address this complexity, we propose a phase-specific therapeutic framework: inhibiting BRD4 during the early hyperinflammatory phase of sepsis may help mitigate excessive inflammation, while strategically activating BRD4 during the subsequent immunosuppressive phase could enhance antimicrobial functions and facilitate effective immune responses. This balanced approach aims to optimize clinical outcomes by harmonizing the need for inflammation with the imperative of microbial clearance.

## Materials and methods

### Ethics Statement

Experiments were performed in compliance with all relevant ethical regulations. Human studies were approved by the Ethics Committee of Fujian Medical University (2023–167), and informed consent was obtained from all participants in written form. Animal studies were approved by the Institutional Animal Care and Use Committee (IACUC) of Fujian Medical University (2023-Y-1132).

Detailed materials and methods can be found in S1 Text.

## Supporting information

**S1 Fig. BRD4 mRNA levels in immune cells (excluding monocytes) show no correlation with serum CRP or PCT levels in septic patients.** (A) Linear mixed-effects model analysis of *BRD4* levels in septic patients across Days 1, 3, and 5. (B) Comparison of *BRD4* mRNA levels in various immune cells between low-CRP and high-CRP septic patients. Data

sources: CMAISE. (C) Comparison of BRD4 mRNA levels in various immune cells between low-PCT and high- PCT septic patients. Data sources: CMAISE. (D) Quantification of *Brd4* mRNA levels in BMDMs at different time points following induction with *E. coli* (left) or *S. aureus* (right) by qRT-PCR.
(TIF)

**S2 Fig. BRD4 expression levels in monocytes/macrophages, but not in other immune cells, negatively correlate with organ damage in sepsis.** (A) Representative TUNEL-stained spleen tissue sections from WT and *Brd4*-CKO mice, 24 hours after sham or CLP surgery. (B) Comparison of *BRD4* mRNA levels in whole blood between septic patients with low and high SOFA scores. Data source: GSE185263. (C) Comparison of *BRD4* mRNA levels in B cells, platelets, and dendritic cells between low-SOFA and high-SOFA septic patients. Data source: SCP548.
(TIF)

**S3 Fig. BRD4 deficiency reduces reactive oxygen species (ROS) and nitric oxide (NO) production in macro-phages.** (A & B) WT and *Brd4*-deficient BMDMs were infected with *E. coli* (A) or *S. aureus* (B). Intracellular ROS levels were measured (n = 3). (C & D) WT and *Brd4*-deficient BMDMs were infected with *E. coli* (C) or *S. aureus* (D). Intracellular NO levels were measured (n = 3).
(TIF)

**S4 Fig. BRD4 deficiency reduces MARCO expression in macrophages.** (A & B) Volcano plot of RNA-seq data comparing WT and *Brd4*-deficient BMDMs challenged with *E. coli* (A) or *S. aureus* (B). Upregulated genes in *Brd4*-deficient BMDMs relative to WT BMDMs are indicated by red dots, downregulated genes by green dots, and genes with no significant change by gray dots. (C) Quantification of MARCO protein levels in WT and *Brd4*-deficient BMDMs after infection with *E. coli* or *S. aureus* for 6 h, as assessed by immunofluorescence.
(TIF)

**S5 Fig. BRD4 promotes the expression of MARCO and MSR1 in macrophages through the activation of NRF2.** (A) WT and *Brd4*-deficient BMDMs were treated with or without LPS (10 ng/mL) for the indicated times. *Marco* mRNA levels were quantified by qRT-PCR. (B) Schematic of the potential binding sites of NRF2 on the murine *Msr1* promoter region, identified by anti-BRD4 ChIP-seq. Data sources: GSE113226. (C) HEK293T cells were co- transfected with Myc-BRD4, with or without Flag-APEX2- NRF2, and treated with or without biotin-4-aminophenol (10 µM). After 30 min, $H_2O_2$ (1 mM) was added for stimulation, followed by immunofluorescence staining and confocal microscopy. (D-F) WT and *Brd4*-deficient BMDMs were treated with or without *E. coli* or *S. aureus* for the indicated times. *Nrf2* (D), *Nqo1* (E) and *Gclc* (F) mRNA levels were quantified by qRT-PCR.
(TIF)

**S6 Fig. The NRF2 inhibitor ML385 abolishes the SFN-mediated enhancement of antibacterial activity in BMDMs.** (A) WT and *Brd4*-deficient BMDMs pre-treated with or without the Sulforaphane (SFN, 10 µM) for 1 h, or with ML385 at two different concentrations: low (5 µM) and high (10 µM) for 12 h, prior to infection with *S. aureus* (MOI = 10) for 4 h. Protein levels of NRF2, MARCO, and MSR1 in cell lysates were assessed by immunoblotting. (B) WT and *Brd4*-deficient BMDMs were pre-treated with or without SFN for 1 h, followed by treatment with GFP-labeled *E. coli* for 1 h. Bacterial phagocytosis was quantified by fluorescence microscopy (n = 3). (C) WT and *Brd4*-deficient BMDMs pre-treated with or without the Sulforaphane (SFN, 10 µM) for 1 h or ML385 at two different concentrations: low (5um) and high (10um) for 12 h were infected with by *S. aureus* (MOI = 10) for 1 h. Bacterial phagocytosis was quantified by fluorescence microscopy (n = 3).
(TIF)

**S7 Fig. ML385 attenuates SFN-mediated protection in septic mice.** WT and *Brd4*-CKO mice were treated with SFN (0.4 mg/kg, i.v.) alone or in combination with ML385 (30 mg/kg, i.p.) and subjected to CLP. (A) Plasma levels of IL-6,

TNF-α, and MCP-1 at 24 h post-CLP (n = 5 per group). (B) Representative H&E-stained sections of liver and lung from the indicated mice at 24 h post-CLP. (C) CFU in blood, peritoneal lavage fluid, lung, spleen, and liver at 24 h post-CLP (n = 5–6 per group).
(TIF)

**S8 Fig. 16S rDNA sequencing analysis of blood from WT mice 24 h after CLP-induced sepsis.** Taxonomic profiling of bacteria at the species level (n = 3).
(TIF)

**S9 Fig. BRD4 deficiency impairs phagocytic and bactericidal functions of neutrophils.** (A & B) Neutrophils isolated from the bone marrow of WT and *Brd4*-CKO mice were analyzed by flow cytometry for purity (A) and by RT-PCR for *Brd4* mRNA levels (B). (C) WT and *Brd4*-deficient neutrophils were infected with GFP-labeled *S. aureus* (MOI = 10) or *E. coli* (MOI = 25) for 1 h. Phagocytosis was quantified by fluorescence microscopy (n = 3). (D) WT and *Brd4*-deficient neutrophils were infected with *S. aureus* (MOI = 10) or *E. coli* (MOI = 50), and bactericidal activity was assessed (n = 3).
(TIF)

**S10 Fig. Reduced peritoneal macrophage populations in *Brd4*-CKO septic mice compared to WT septic mice.** (A & B) Representative flow cytometry plots showing peritoneal macrophages (A) and neutrophils (B) from WT and *Brd4*-CKO mice, 24 hours after sham or CLP surgery (n = 5).
(TIF)

**S11 Fig. BRD4 deficiency does not affect KEAP1 expression in macrophages.** WT and *Brd4*-deficient BMDMs were infected with or without *E. coli* or *S. aureus* for the indicated times. *Keap1* mRNA levels were quantified by qRT-PCR.
(TIF)

**S12 Fig. Genome-wide co-occupancy of BRD4 and NRF2 in BMDMs.** (A) Heatmaps depicting BRD4 and NRF2 binding signals in BMDMs based on published ChIP-seq datasets (BRD4: GSE113226; NRF2: DRA003771). Color scales indicate signal intensity. (B) Venn diagram showing genomic co-localization of BRD4 and NRF2. (C) Gene Ontology (GO) analysis of genes co-occupied by BRD4 and NRF2. (D) KEGG pathway analysis of genes co-occupied by BRD4 and NRF2.
(TIF)

**S13 Fig. BRD4's inhibitory effect on NRF2 ubiquitination is not mediated through the recruitment of deubiquitinases, or modulation of Cullin3 activity.** (A) WT and *Brd4*-knockout HEK293T cells expressing Myc-KEAP1 and immunoprecipitated with anti-Myc beads and analyzed by immunoblotting. (B & C) WT and Brd4-knockout HEK293T cells co-expressing HA-USP7 and Flag-NRF2 (B), HA-USP11 and Flag-NRF2 (C), were immunoprecipitated using anti-HA beads. The interaction was detected by immunoblotting. (D) WT and *Brd4*-knockout HEK293T cells co-transfected with Myc-DUB3 and Flag-NRF2 and cell lysates were immunoprecipitated with anti-Myc beads and analyzed by immunoblotting.
(TIF)

**S1 Table. Multivariable analysis of *BRD4* or *MARCO* expression in sepsis and healthy controls (adjusted for age and sex).**
(DOCX)

**S2 Table. Reagents and Resources.**
(DOCX)

**S3 Table. Primers used for PCR.**
(DOCX)

**S1 Text. Detailed materials and methods.**
(DOCX)

## Acknowledgments

We thank Dr. Xi Mo (Shanghai Jiao Tong University) for critical reading of this manuscript.

## Author contributions

**Conceptualization:** Jinfeng Hu, Xiaopei Shen, Xiangming Hu.

**Data curation:** Jinfeng Hu, Xuming Gao, Guo Li, Xiaoxin He, Yanting Ke, Suhong Yu, Xiaopei Shen.

**Formal analysis:** Jinfeng Hu, Xuming Gao, Guo Li, Xiaoxin He, Yanting Ke.

**Funding acquisition:** Jinfeng Hu, Xuming Gao, Xiangming Hu.

**Investigation:** Jinfeng Hu, Xuming Gao, Guo Li, Xiaoxin He, Duozhi Pang.

**Methodology:** Minting Jiang, Shuping Zheng.

**Resources:** Zhongheng Zhang, Mingrui Lin, Dun Pan.

**Software:** Yanting Ke, Xiaopei Shen.

**Supervision:** Jinfeng Hu, Xiaopei Shen, Xiangming Hu.

**Validation:** Jinfeng Hu, Guo Li, Duozhi Pang, Zhen Lin, Cailian Xie, Xiaoting Chen.

**Writing – original draft:** Jinfeng Hu, Xiaopei Shen, Xiangming Hu.

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
