## [Decision Letter · Decision Letter 0]

14 Dec 2025

PPATHOGENS-D-25-02280

BRD4 modulates antimicrobial defense via non-canonical NRF2 activation in macrophages to confer protection against sepsis

PLOS Pathogens

Dear Dr. Hu,

Thank you for submitting your manuscript to PLOS Pathogens. After careful consideration, we feel that it has merit but does not fully meet PLOS Pathogens's publication criteria as it currently stands. Therefore, we invite you to submit a revised version of the manuscript that addresses the points raised during the review process.

We look forward to receiving your revised manuscript.

Kind regards,

Vasundhra Bhandari

Academic Editor

PLOS Pathogens

Thomas Guillard

Section Editor

PLOS Pathogens Sumita Bhaduri-McIntosh

Editor-in-Chief

PLOS Pathogens

orcid.org/0000-0003-2946-9497

Michael Malim

Editor-in-Chief

PLOS Pathogens

orcid.org/0000-0002-7699-2064

**Journal Requirements:**

1) Thank you for including an Ethics Statement for your study. Please state whether the consent obtained is verbal or written.  2) Please upload all main figures as separate Figure files in .tif or .eps format. For more information about how to convert and format your figure files please see our guidelines:  https://journals.plos.org/plospathogens/s/figures  3) We have noticed that you have uploaded Supporting Information files, but you have not included a complete list of legends. Please add a full list of legends for your Supporting Information files (Raw Data of Figs-Sepsis-091625.xlsx and Raw data of WB_Sepsis-ok.pdf) after the references list. 4) Some material included in your submission may be copyrighted. According to PLOSu2019s copyright policy, authors who use figures or other material (e.g., graphics, clipart, maps) from another author or copyright holder must demonstrate or obtain permission to publish this material under the Creative Commons Attribution 4.0 International (CC BY 4.0) License used by PLOS journals. Please closely review the details of PLOSu2019s copyright requirements here: PLOS Licenses and Copyright. If you need to request permissions from a copyright holder, you may use PLOS's Copyright Content Permission form. Please respond directly to this email and provide any known details concerning your material's license terms and permissions required for reuse, even if you have not yet obtained copyright permissions or are unsure of your material's copyright compatibility. Once you have responded and addressed all other outstanding technical requirements, you may resubmit your manuscript within Editorial Manager.  Potential Copyright Issues: i) Figures 2B, 2D, 2G, 2J, 2K, 2L, 8D, 8E, 8F, 9, and S2A. Please confirm whether you drew the images / clip-art within the figure panels by hand. If you did not draw the images, please provide (a) a link to the source of the images or icons and their license / terms of use; or (b) written permission from the copyright holder to publish the images or icons under our CC BY 4.0 license. Alternatively, you may replace the images with open source alternatives. See these open source resources you may use to replace images / clip-art: - https://commons.wikimedia.org -
https://openclipart.org/. ii) Figure 5 : Thank you for stating that "The diagram was adapted from the GEO Datasets of the National Center for Biotechnology Information (GSE113226). " Please provide us with a direct link to the original image source and provide a link to the terms of use / license information.

Note : If the figure is adapted from a copyrighted source, please provide written permission from the copyright holder to publish this under our CC-BY 4.0 license, or remove the figure / replace the image. Please note we do not recommend using standard request forms available on Publishers' websites, as they grant single use rather than republication under an open access license  5) In the online submission form, you indicated that "The data are available from the corresponding author upon request;" however, you uploaded raw data as supporting information files.

Please provide a complete Data Availability Statement in the online submission form, ensuring you include all necessary access information. If your research concerns only data provided within your submission, please write "All data are in the manuscript and/or supporting information files" as your Data Availability Statement.

If your research concerns data from external sources, please amend your Data Availability Statement to include the full link to the data.  6) Your current Financial Disclosure states, "The author(s) received no specific funding for this work. However, your funding information on the submission form indicates receiving a fund. Please ensure that the funders and grant numbers match between the Financial Disclosure field and the Funding Information tab in your submission form. Note that the funders must be provided in the same order in both places as well. Please amend your detailed Financial Disclosure statement. This is published with the article. It must therefore be completed in full sentences and contain the exact wording you wish to be published.1) Please clarify all sources of financial support for your study. List the grants, grant numbers, and organizations that funded your study, including funding received from your institution. Please note that suppliers of material support, including research materials, should be recognized in the Acknowledgements section rather than in the Financial Disclosure

2) State the initials, alongside each funding source, of each author to receive each grant. For example: "This work was supported by the National Institutes of Health (####### to AM; ###### to CJ) and the National Science Foundation (###### to AM)."

3) State what role the funders took in the study. If the funders had no role in your study, please state: "The funders had no role in study design, data collection and analysis, decision to publish, or preparation of the manuscript."

4) If any authors received a salary from any of your funders, please state which authors and which funders.. 7) Please revise your current Competing Interest statement to the standard "The authors have declared that no competing interests exist."

**Reviewers' Comments:**

Reviewer's Responses to Questions

**Part I - Summary**

Reviewer #1: This manuscript provides an important conceptual advance by positioning BRD4–NRF2 as a central regulatory axis in innate immunity during sepsis. The integration of mouse genetics, cellular functional assays, and human data is commendable. With additional experiments—particularly verifying NRF2 dependence, endogenous BRD4–NRF2 interaction, and proper statistical control—this work would be substantially strengthened. I recommend major revision, but I am confident that after revision, this study will have high impact and broad relevance.

Reviewer #2: In this manuscript, Hu and colleagues studied the pathogenic role of BRD4, an epigenetic regulator, in sepsis, a life-threatening condition characterized by dysregulated immune

responses and high mortality, with a focus on the relevance of BRD4 in monocytes/macrophages. The claimed myeloid-specific function of BRD4 was supported by the evidence from human patients and murine CLP models. Mechanistically, the interaction between BRD4 and NRF2, which disrupts the NRF2-KEAP1 complex thus enhancing NRF2 stability and nuclear translocation and upregulating scavenger receptors essential for bacterial clearance. The role of BRD4 in monocytes/macrophages and sepsis was explored previously, which dampens the novelty of current study; however, the data presented, particularly those in the mouse CLP model would be of interest to the field.

**Part II – Major Issues: Key Experiments Required for Acceptance**

Reviewer #1: This study identifies the BRD4–NRF2 axis as a novel regulatory pathway controlling macrophage antimicrobial defense in sepsis—an insightful connection between epigenetic regulation and host immunity. Using both human and murine data, the authors convincingly show that BRD4 expression decreases in monocytes/macrophages and correlates with disease severity. The use of myeloid-specific Brd4 knockout mice demonstrates that BRD4 loss leads to impaired bacterial clearance and higher mortality. The mechanistic link between BRD4-mediated NRF2 stabilization, scavenger receptor induction, and the rescue effect by sulforaphane (SFN) suggests important therapeutic potential. Overall, the manuscript is solid and of high impact, suitable for PLOS Pathogens, but several aspects require further strengthening.

1. The causal relationship between BRD4 and NRF2 activation needs to be demonstrated more rigorously. The authors conclude that BRD4 stabilizes NRF2, thereby inducing MARCO/MSR1 expression, but they have not performed epistasis analyses in vivo (e.g., double knockout Brd4^f/f; LysM-Cre × Nrf2^f/f, or NRF2 overexpression rescue). To confirm that the proposed BRD4→NRF2 signaling axis is causal, additional experiments are necessary. The authors should test whether the rescue effect of SFN depends on NRF2—using either pharmacologic inhibition (ML385) in vitro or myeloid-specific Nrf2 deletion in vivo. These experiments would establish the directionality and NRF2-dependence of BRD4’s effects.

2. The direct interaction between BRD4 and NRF2 should be more conclusively demonstrated. The current co-immunoprecipitation data rely largely on overexpression in 293T cells. Endogenous BRD4–NRF2 binding should be confirmed in primary macrophages using co-IP or proximity ligation assays (PLA). To support the proposed KEAP1 competition mechanism, in vitro competition assays with recombinant proteins (pull-down or SPR) would be highly informative. In addition, genome-wide analyses (ChIP-seq or CUT&RUN) for NRF2 and BRD4 co-occupancy would strengthen the evidence for cooperative transcriptional regulation.

3. The SFN rescue experiments are compelling but need to rule out off-target effects. Because SFN affects multiple pathways, the authors should clarify whether its protective effects are strictly NRF2-dependent. Combining SFN with NRF2 inhibition, and testing different doses or treatment timing, would improve translational relevance and mechanistic specificity. Although SFN improved survival in both WT and Brd4-CKO mice, this does not by itself prove NRF2 exclusivity—clarifying this would strengthen the conclusion.

4. Concerning the LysM-Cre model, the manuscript should acknowledge its known limitations. LysM-Cre drives recombination in both macrophages and neutrophils; thus, the phenotype might partly reflect altered neutrophil function. Assessing neutrophil phagocytosis, bactericidal activity, or NETosis—or reproducing key experiments using a macrophage-specific driver (e.g., Cx3cr1-CreER)—would help to exclude this confounding factor.

5. The human data are of high translational value but require better statistical control. Multivariate analyses adjusting for age, sex, infection source, comorbidities, and treatment variables should be presented to confirm the independent association between BRD4 (and NRF2) expression and clinical severity. Receiver operating characteristic (ROC) analysis for prognostic performance would also add rigor. Furthermore, sampling timing and disease stage heterogeneity should be clarified, and longitudinal analysis of expression dynamics would be valuable.

6. Mechanistically, the authors could explore how BRD4 inhibits NRF2 ubiquitination beyond KEAP1 competition—whether it recruits deubiquitinases or influences Cullin3 E3 ligase activity.

7. In the discussion, the authors should elaborate on the dual nature of BRD4—as both an enhancer of inflammatory gene expression and a protector of antimicrobial function. Presenting a phase-specific therapeutic framework (inhibiting BRD4 in early hyperinflammation, activating it during the immunosuppressive phase) would increase clinical relevance. Additionally, potential risks of sustained NRF2 activation (tumor promotion, metabolic effects) should be briefly mentioned.

Reviewer #2: Figure 3A, it is not clear the bacterial burden was measured under anaerobic or aerobic culture condition? If the colony-formed Gram-negative and -positive bacteria could be isolated and identified, would they be more relevant species than E. coli and S. aureus to assess the myeloid-specific impact of BRD4 in bacterial clearance and sepsis, as in Figure 3?

Myeloid-specific Brd4 knockout mice was previously shown resistant to LPS-induced sepsis, while the current study demonstrated that Brd4 conditional knockout mice were more susceptible to CLP-induced polymicrobial sepsis (ref 21). Such phenotype difference should be discussed in depth, in particular, the identification of the recruitment of Brd4 to the promoters of Marco and Msr1 was based on the LPS treatment data (Supplementary Figure 5 and Figure 5A).

What were the transcription levels of Brd4 in Figure 1 J and K? It would be necessary to ascertain whether the decreased levels of BRD4 in later time points could result from the protein degradation.

**Part III – Minor Issues: Editorial and Data Presentation Modifications**

Reviewer #1: (No Response)

Reviewer #2: (No Response)

PLOS authors have the option to publish the peer review history of their article (what does this mean?). If published, this will include your full peer review and any attached files.

Reviewer #1: No

Reviewer #2: **Yes:** Fengyi Wan

**Figure resubmission:**

While revising your submission, we strongly recommend that you use PLOS’s NAAS tool (https://ngplosjournals.pagemajik.ai/artanalysis) to test your figure files. NAAS can convert your figure files to the TIFF file type and meet basic requirements (such as print size, resolution), or provide you with a report on issues that do not meet our requirements and that NAAS cannot fix. After uploading your figures to PLOS’s NAAS tool - https://ngplosjournals.pagemajik.ai/artanalysis, NAAS will process the files provided and display the results in the "Uploaded Files" section of the page as the processing is complete. If the uploaded figures meet our requirements (or NAAS is able to fix the files to meet our requirements), the figure will be marked as "fixed" above. If NAAS is unable to fix the files, a red "failed" label will appear above. When NAAS has confirmed that the figure files meet our requirements, please download the file via the download option, and include these NAAS processed figure files when submitting your revised manuscript.
---

## [Decision Letter · Decision Letter 1]

18 Mar 2026

PPATHOGENS-D-25-02280R1

BRD4 modulates antimicrobial defense via non-canonical NRF2 activation in macrophages to confer protection against sepsis

PLOS Pathogens

Dear Dr. Hu,

Thank you for submitting your manuscript to PLOS Pathogens. After careful consideration, we feel that it has merit but does not fully meet PLOS Pathogens's publication criteria as it currently stands. Therefore, we invite you to submit a revised version of the manuscript that addresses the points raised during the review process.

We look forward to receiving your revised manuscript.

Kind regards,

Vasundhra Bhandari

Academic Editor

PLOS Pathogens

Thomas Guillard

Section Editor

PLOS Pathogens

Sumita Bhaduri-McIntosh

Editor-in-Chief

PLOS Pathogens

orcid.org/0000-0003-2946-9497

Michael Malim

Editor-in-Chief

PLOS Pathogens

orcid.org/0000-0002-7699-2064

**Reviewers' Comments:**

Reviewer's Responses to Questions

**Part I - Summary**

Reviewer #1: The authors have carefully addressed the major concerns raised in the previous round of review and have performed several important additional experiments that substantially strengthen the manuscript. In particular, the newly added NRF2 reconstitution experiments in Brd4 conditional knockout mice, the endogenous BRD4–NRF2 interaction analysis in primary macrophages, and the NRF2 inhibitor (ML385) experiments confirming the specificity of SFN-mediated rescue significantly improve the mechanistic rigor of the study. These new data provide convincing support for the proposed BRD4–NRF2 signaling axis in regulating macrophage antimicrobial function during sepsis. The authors have also made reasonable efforts to address the concerns regarding the LysM-Cre model by examining neutrophil antimicrobial functions, which adds valuable insight into the broader role of BRD4 in myeloid innate immunity.

Reviewer #2: The authors have addressed the concerns and critiques in a satisfying manner. The revised verson of the manuscript with newly added experimental data and extensive revisions is suitable for publication.

**Part II – Major Issues: Key Experiments Required for Acceptance**

Reviewer #1: (No Response)

Reviewer #2: Not applicable.

**Part III – Minor Issues: Editorial and Data Presentation Modifications**

Reviewer #1: Overall, the revised manuscript represents a substantial improvement and now provides a coherent mechanistic framework linking BRD4 to NRF2-dependent antimicrobial responses.

However, a few minor issues remain that should be clarified before publication.

1. The ChIP-seq analysis supporting BRD4–NRF2 co-occupancy relies on publicly available datasets. While this analysis provides useful bioinformatic support, the authors may clarify this limitation in the manuscript and discuss potential context-dependent differences between experimental systems.

2. In the KEAP1 competition assays, a BRD4 deletion construct (BRD4-PID) was used instead of full-length BRD4. The authors may briefly discuss whether additional domains of BRD4 could influence NRF2 binding in vivo.

3. The discussion of the LysM-Cre model would benefit from a brief statement acknowledging that both macrophages and neutrophils contribute to the observed phenotype, as suggested by the newly added neutrophil functional assays.

Reviewer #2: Not applicable.

PLOS authors have the option to publish the peer review history of their article (what does this mean?). If published, this will include your full peer review and any attached files.

Reviewer #1: No

Reviewer #2: No

**Figure resubmission:**
---

## [Decision Letter · Decision Letter 2]

22 Apr 2026

Dear Prof. Hu,

We are pleased to inform you that your manuscript 'BRD4 modulates antimicrobial defense via non-canonical NRF2 activation in macrophages to confer protection against sepsis' has been provisionally accepted for publication in PLOS Pathogens.

Best regards,

Vasundhra Bhandari

Academic Editor

PLOS Pathogens

Thomas Guillard

Section Editor

PLOS Pathogens

Sumita Bhaduri-McIntosh

Editor-in-Chief

PLOS Pathogens

orcid.org/0000-0003-2946-9497

Michael Malim

Editor-in-Chief

PLOS Pathogens

orcid.org/0000-0002-7699-2064

Reviewer Comments (if any, and for reference):

Reviewer's Responses to Questions

**Part I - Summary**

Reviewer #1: (No Response)

**Part II – Major Issues: Key Experiments Required for Acceptance**

Reviewer #1: (No Response)

**Part III – Minor Issues: Editorial and Data Presentation Modifications**

Reviewer #1: (No Response)

PLOS authors have the option to publish the peer review history of their article (what does this mean?). If published, this will include your full peer review and any attached files.

Reviewer #1: **Yes:** Shigeki Nakamura

---

## [Editor Report · Acceptance letter]

Dear Prof. Hu,

We are delighted to inform you that your manuscript, "BRD4 modulates antimicrobial defense via non-canonical NRF2 activation in macrophages to confer protection against sepsis," has been formally accepted for publication in PLOS Pathogens.

Best regards,

Sumita Bhaduri-McIntosh

Editor-in-Chief

PLOS Pathogens

orcid.org/0000-0003-2946-9497

Michael Malim

Editor-in-Chief

PLOS Pathogens

orcid.org/0000-0002-7699-2064